# Peripheral blood transcriptome profiling enables monitoring disease progression in dystrophic mice and patients

Mirko Signorelli[1,*] ID, Mitra Ebrahimpoor[1], Olga Veth[2], Kristina Hettne[2] ID, Nisha Verwey[2], Raquel García-Rodríguez[2], Christa L Tanganyika-deWinter[2], Luz B Lopez Hernandez[3,4], Rosa Escobar Cedillo[5], Benjamín Gómez Díaz[5], Olafur T Magnusson[6], Hailiang Mei[7], Roula Tsonaka[1], Annemieke Aartsma-Rus[2] & Pietro Spitali[2,**] ID

## Abstract

DMD is a rare disorder characterized by progressive muscle degeneration and premature death. Therapy development is delayed by difficulties to monitor efficacy non-invasively in clinical trials. In this study, we used RNA-sequencing to describe the pathophysiological changes in skeletal muscle of 3 dystrophic mouse models. We show how dystrophic changes in muscle are reflected in blood by analyzing paired muscle and blood samples. Analysis of repeated blood measurements followed the dystrophic signature at five equally spaced time points over a period of seven months. Treatment with two antisense drugs harboring different levels of dystrophin recovery identified genes associated with safety and efficacy. Evaluation of the blood gene expression in a cohort of DMD patients enabled the comparison between preclinical models and patients, and the identification of genes associated with physical performance, treatment with corticosteroids and body measures. The presented results provide evidence that blood RNA-sequencing can serve as a tool to evaluate disease progression in dystrophic mice and patients, as well as to monitor response to (dystrophin-restoring) therapies in preclinical drug development and in clinical trials.

**Keywords** biomarkers; Duchenne muscular dystrophy; dystrophinopathies; RNA-seq

**Subject Categories** Biomarkers; Musculoskeletal System

## Introduction

Duchenne muscular dystrophy (DMD) is a pediatric disorder characterized by severe disease progression and premature death (Mercuri & Muntoni, 2013). It is the most common form of muscular dystrophy, with an incidence of 1 in 5000 male births. Even with good multidisciplinary care, DMD patients show delayed motor milestones, loss of ambulation in the teens, inability to self-feed in the twenties, and reduced life expectancy (Birnkrant *et al*, 2018b). Most deaths are observed in the 30s due to cardio-pulmonary complications (Birnkrant *et al*, 2018a). DMD is caused by lack of dystrophin due to mutations in the *DMD* gene, one of the biggest genes in the genome spanning more than 2.2 Mb. Analysis of patients' muscle biopsies and of skeletal muscles models has led to the identification of the molecular cascade of events leading to muscle damage, inflammation, and fibrosis and adiposis formation characteristic of DMD (Bakay *et al*, 2002; Coenen-Stass *et al*, 2018; Dowling *et al*, 2019). The knowledge of the disease pathogenesis triggered the development of therapeutic agents targeting a number of affected molecular targets, such as NF-κB, IGF-1, and myostatin (Secco *et al*, 2013; Heier *et al*, 2019; Hammers *et al*, 2019). However, these trials have often been disappointing (e.g., blockade of myostatin signaling with different biologics (Wagner *et al*, 2020) and the NF-κB inhibitor edasalonexent), with drugs failing due to safety issues, reduced drug potency, and insufficient power in the study design (Goemans *et al*, 2018; Verhaart & Aartsma-Rus, 2019). Indeed, the drug development process would be facilitated by more knowledge connecting the biology of the disease with clinical progression, and by the ability to objectively monitor clinically relevant changes in a non-invasive manner.

1   Department of Biomedical Data Sciences, Leiden University Medical Center, Leiden, The Netherlands
2   Department of Human Genetics, Leiden University Medical Center, Leiden, The Netherlands
3   Departamento de Medicina Genómica, Universidad Autónoma de Guadalajara, Guadalajara, Mexico
4   Centro Médico Nacional "20 de Noviembre", ISSSTE, Ciudad de México, Mexico
5   Sociedad Mexicana de la Distrofia Muscular A.C INR-LGII, Ciudad de México, Mexico
6   deCODE genetics/Amgen, Reykjavik, Iceland
7   Sequencing Analysis Support Core, Leiden University Medical Center, Leiden, The Netherlands
    *Corresponding author. Tel: +31 715268568; Email: m.signorelli@lumc.nl
    **Corresponding author. Tel: +31 715269437; Email: p.spitali@lumc.nl

Historically, biomolecular evidence was derived from the analysis of the affected muscles obtained from biopsy material and animal models (Turk *et al*, 2006). Lately, more studies are reporting molecular changes in more accessible sample matrices such as plasma, serum, and urine (Ayoglu *et al*, 2014; Hathout *et al*, 2015; Robertson *et al*, 2017; Tsonaka *et al*, 2020). The overall goal of these studies is to provide molecular evidence related to the biology of the affected tissue and connected clinical parameters, while reducing the invasiveness of the sampling procedure. This is particularly important for pediatric patients such as DMD patients, who are typically enrolled in clinical trials when they are between 5 and 8 years old. Biobanking and natural history studies have made it possible to use blood samples to identify direct associations between serum biomarkers and functional scales, and to improve prediction of disease milestones such as loss of ambulation (Strandberg *et al*, 2020; Signorelli *et al*, 2020a; preprint: Signorelli *et al*, 2021). While these studies show initial evidence that molecular biomarkers can be used to anticipate clinical outcomes in patients, it is often not trivial to relate blood biomarkers to the ongoing pathophysiology in muscle under physiologic conditions and during exposure to therapeutic agents (Al-Khalili Szigyarto & Spitali, 2018).

Establishing clear relationships between muscle pathophysiology and peripheral biomarkers would make it possible to describe changes into the affected tissue without having to resort to muscle biopsies. Connecting muscle-specific drug effects and blood measurements could facilitate pharmacodynamic studies. Building clear reasoning of how the expected changes in muscle biology relate to blood observations and patient performance would enable a smoother regulatory process.

To achieve integration of muscle and blood biological signatures, we sought to compare gene expression in paired muscle and blood samples by RNA-sequencing (RNA-seq). The impossibility to obtain paired blood and muscle samples in natural history studies and clinical trials led us to design a study involving animal models where we could link muscle and blood gene expression in three different mouse models of DMD (Fig 1). The presented data show a strong gene expression signature in the muscle of dystrophic mice, which is conserved across the three dystrophic groups. Analysis of muscle and blood data showed how muscle gene expression can partly be studied by monitoring gene expression in blood.

We further explored how the blood signature is preserved over time through a 7 months long observational study. This enabled us to show how the dystrophic signature in mice is maintained over a study duration exceeding the duration for most interventional studies in animals. In addition, we restored dystrophin with two antisense oligonucleotide drugs and evaluated the extent to which the dystrophic signature can be restored following treatment.

Finally, we studied blood gene expression in a cohort of DMD patients, observing large deviations from healthy controls and identifying genes significantly associated with steroid treatment, patients physical characteristics, and their performance as measured by multiple scales. Taken together, the results presented in this article provide a large set of observations able to link blood biomarkers to patients performance, longitudinal trajectories, and muscle pathophysiology, enriching the clinical trial toolkit for drug developers and investigators.

# Results

## Analysis of gene expression in muscle tissue identifies a large number of dysregulated genes in various *mdx* groups

Confirming previous findings, preliminary exploration of the MoMus dataset by PCA pointed out a clear separation between WT and dystrophic (*mdx*, *mdx++*, *mdx+−*) mice in the space defined by the first two principal components (Fig 2A). Hypothesis testing of differential expression between groups showed that a large proportion (42%) of the genes considered in the analysis was differentially expressed (FDR < 0.05, Fig 2B) between WT and *mdx* mice (Dataset EV2, Tab 1_WT_mdx). Of the 4660 genes differentially expressed between WT and *mdx* mice, 4246 were also differentially expressed between WT and *mdx++* mice (Fig 2C and Dataset EV2, Tab 2_WT_mdx++), and 3874 between WT and *mdx+−* mice (Fig 2D and Dataset EV2, Tab 3_WT_mdx+−). In all three paired comparisons, the *Dmd* gene (Fig 2E) was the gene for which evidence of differential expression was stronger (adjusted p-value < $10^{-13}$ in each of the 3 comparisons).

Differential expression was highly conserved across the 3 dystrophic mouse models, with 8 genes (*Dmd*, *Prune2*, *Slc15a5*, *Efcab6*, *Ces1d*, *Dach2*, *6330416G13Rik,* and *Ephx2*) shared by the three lists of top 10 differentially expressed genes for *mdx*, *mdx++,* and *mdx+−* mice (Dataset EV2). For example, *Prune2* (Fig 2F) was strongly upregulated in *mdx*, *mdx++*, and *mdx+−* mice (adjusted p-value < $10^{-13}$ in each of the 3 comparisons).

Comparison of the log-fold changes (logFCs) between each *mdx* mouse group and WT mice further highlighted a strong similarity between the gene expression profiles of *mdx*, *mdx++,* and *mdx+−* mice (Figs 2H–J), with largely concordant fold changes across the 3 mouse models. Comparison of *mdx++* and *mdx+−* mice, previously reported to show differences in disease severity (van Putten *et al*, 2012), did not lead to the identification of any differentially expressed gene (Fig 2G and Dataset EV2, Tab 4_mdx++_mdx+−). Ingenuity pathway analysis of all dystrophic groups versus WT animals showed down-regulation of metabolic pathways and increased immune and inflammation pathways (Fig 2K).

## Longitudinal analysis of blood samples finds stronger evidence of differential expression between WT and *mdx* mice at weeks 6, 12, and 18

Statistical modeling of the longitudinal trajectories of genes in the MoLong dataset allowed to identify 1532 genes as differentially expressed between WT and *mdx* mice (FDR < 0.05, Fig 3A and Dataset EV3, Tab 1_WT_mdx). For these genes, the comparison of the logFCs at different weeks highlighted the presence of both upregulated and downregulated genes at each week, with a prevalence of genes upregulated in *mdx* mice at weeks 6, 12, and 24 (Fig 3B).

Hypothesis testing of differences at each week showed that most of those genes (859 out of 1532) were significantly different at week 6 (FDR < 0.05, Dataset EV3, Tab 1_WT_mdx), and that the strength of group differences was reduced in later weeks, with 416 significant genes at week 12, 459 at week 18, 134 at week 24, and 83 at week 30. A study of the overlap between the lists of significant genes at

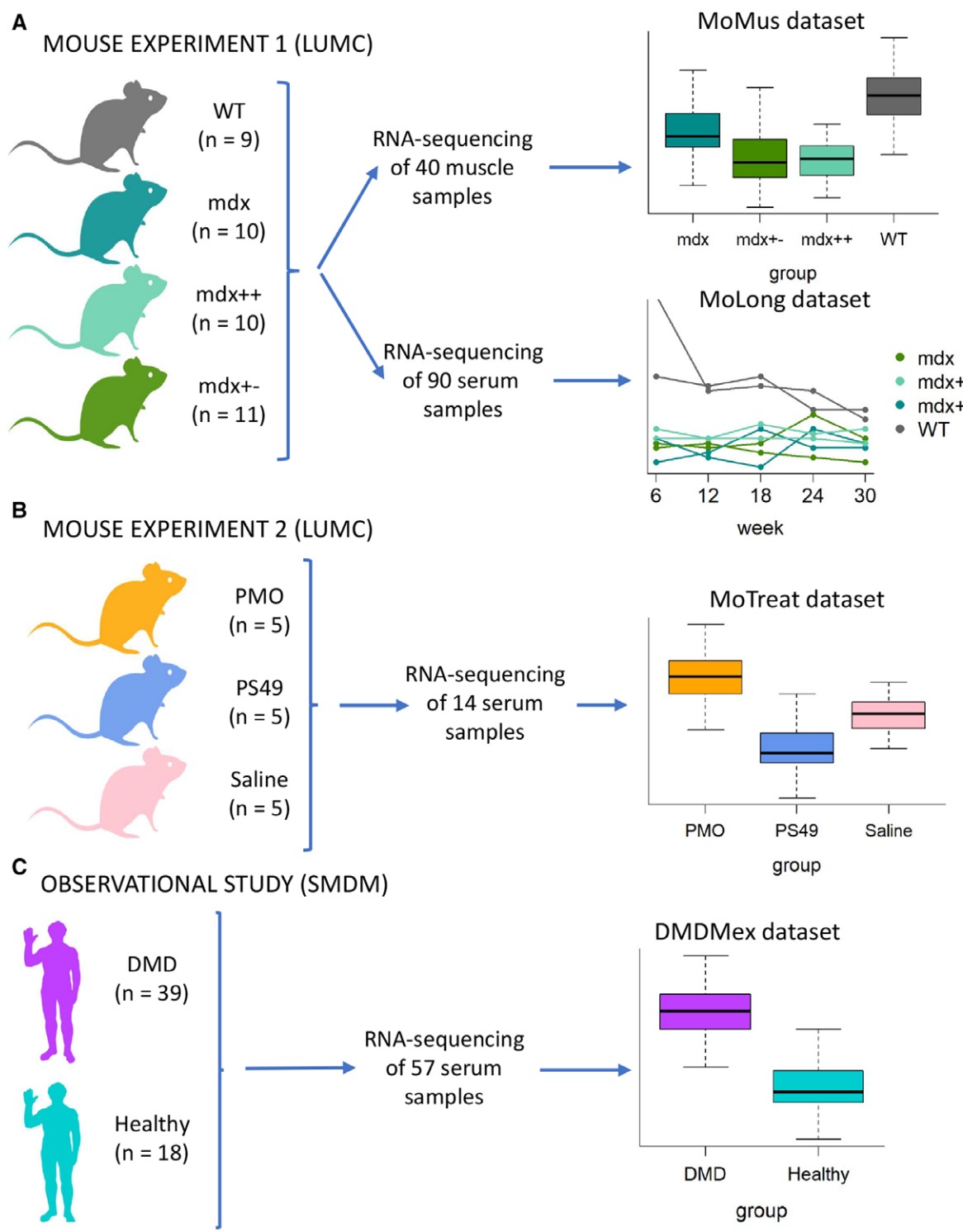

**Figure 1. Overview of the experiments and observational study presented in this article.**

A The first mouse experiment performed at LUMC was set up to compare gene expression profiles of healthy (WT) and dystrophic (mdx, mdx++ and mdx+−) mice. It involved 40 mice and resulted in the collection of two datasets, one with cross-sectional RNA-sequencing in muscle (MoMus) and the other with longitudinal RNA-sequencing measurements in blood (MoLong).

B The second mouse experiment performed at LUMC aimed to assess the effect of dystrophin restoration by two antisense oligonucleotides (PMO and PS49) on blood gene expression. It involved 15 dystrophic mice and led to the generation of a cross-sectional dataset (MoTreat) with blood RNA-sequencing data.

C The observational study carried out at SMDM involved 18 healthy subjects and 39 DMD patients. It resulted in the collection of a cross-sectional dataset (DMDMex) with blood RNA-sequencing data.

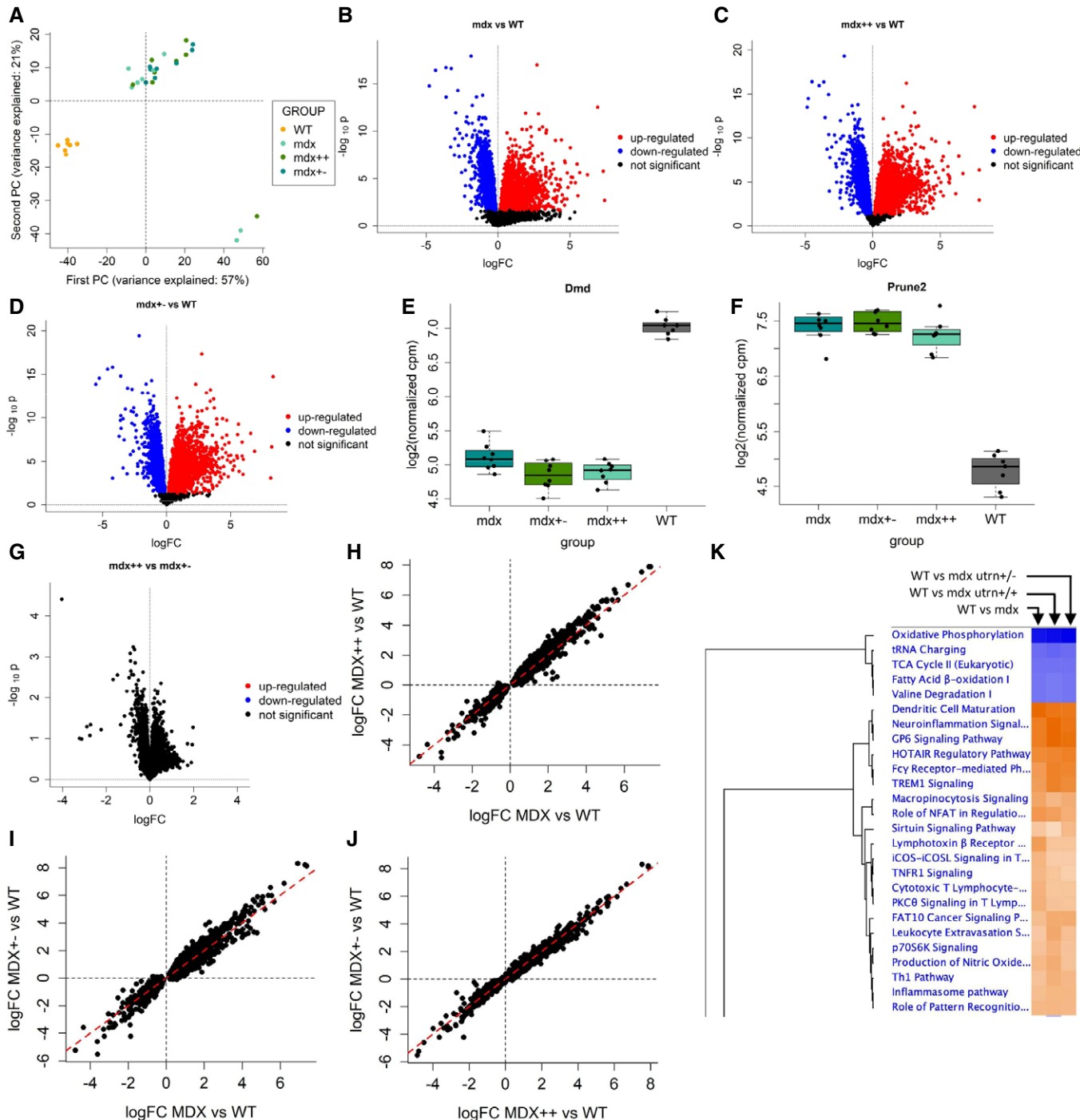

**Figure 2. Results of the analysis of muscle tissue (MoMus dataset).**

A   Principal component analysis of the MoMus RNA-seq counts.

B   Volcano plot for the *F* test on differential expression between WT and mdx mice.

C   Volcano plot for the *F* test on differential expression between WT and mdx++ mice.

D   Volcano plot for the *F* test on differential expression between WT and mdx+− mice.

E, F   Boxplot comparing the distribution of the *Dmd* and *Prune2* genes in the WT, mdx, mdx++, and mdx+− mice groups. The central band of the boxplot denotes the median; the box denotes the first and third quartile; the whiskers are computed as min(max(x), Q3 + 1.5 * IQR) and max(min(x), Q1 − 1.5 * IQR), where min(x) and max(x) denote the minimum and maximum of the distribution, Q1 and Q3 the first and third quartile, and IQR the interquartile range.

G   Volcano plot for the *F* tests on differential expression between mdx++ and mdx+− mice.

H–J   Scatter plots comparing the estimated logFCs across mice groups. Panel H: mdx vs. mdx++. Panel I: mdx vs mdx+−. Panel J: mdx++ vs. mdx+−.

K   Heatmap obtained by Ingenuity Pathway Analysis showing the most affected pathways across mdx, mdx++, and mdx+− mice.

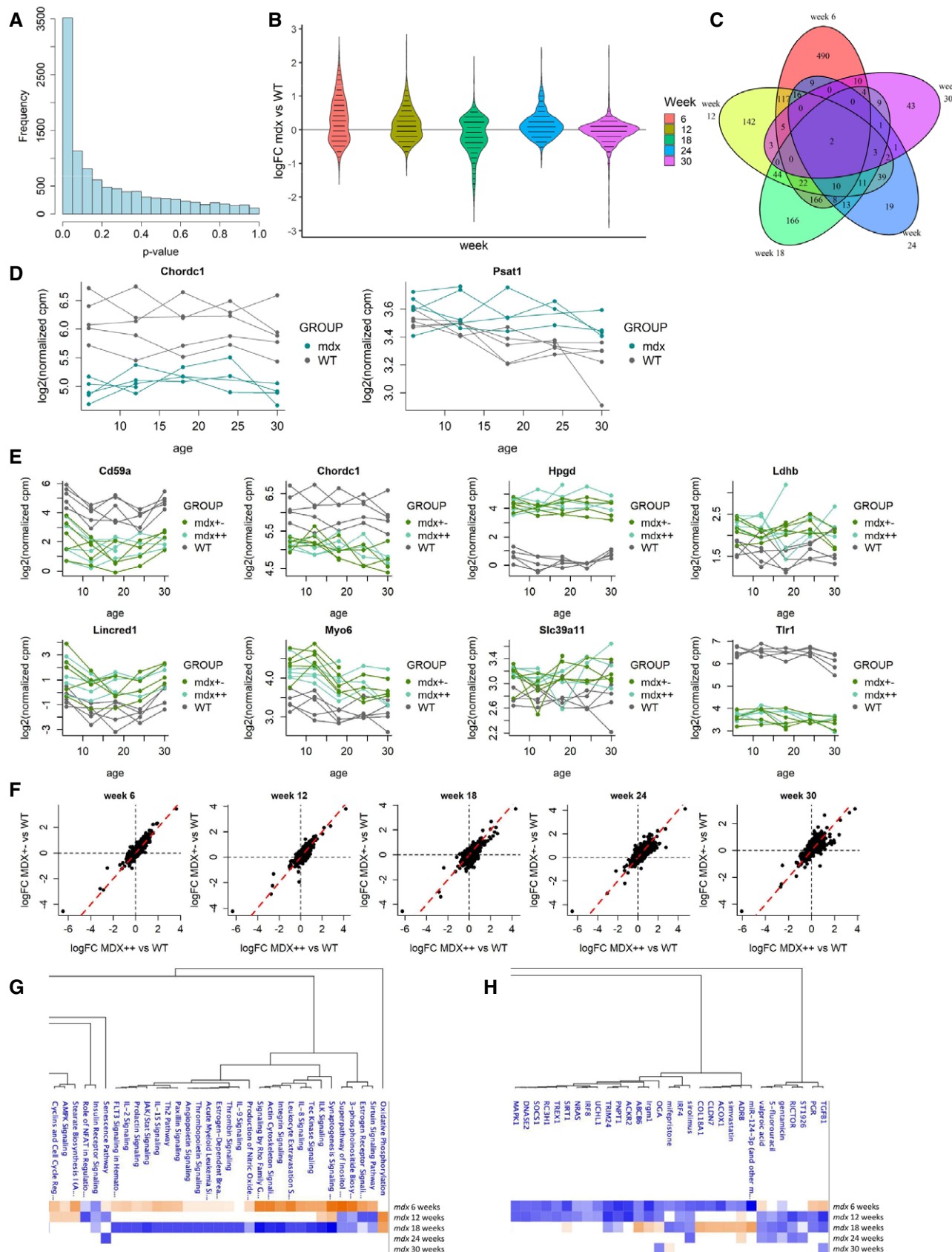

**Figure 3.**

◀

**Figure 3. Results of the analysis of blood tissue (MoLong dataset).**

A    Histogram of the likelihood ratio p-values for the test on differences between WT and *mdx* at any time point.
B    Distribution of the estimated logFCs for *mdx* versus WT mice at weeks 6, 12, 18, 24, and 30 for the 1,532 genes with significant differences between WT and *mdx*.
C    Venn diagram showing the overlap between the sets of genes with significant differences between WT and *mdx* mice at different weeks.
D    Trajectory plots comparing the trajectories of *Chordc1* and *Psat1* in WT and *mdx* mice.
E    Trajectory plots comparing the trajectories of *Cd59a, Chordc1, Hpgd, Ldhb, Lincred1, Myo6, Slc39a11,* and *Tlr1* in WT, *mdx++*, and *mdx+−* mice.
F    Scatter plots comparing the estimated logFC of *mdx++* vs WT (x-axis) and of *mdx+−* vs. WT (y-axis) at weeks 6, 12, 18, 24, and 30.
G, H  Pathway analysis (G) and upstream regulator analysis (H) performed by Ingenuity Pathway Analysis of the comparison between *mdx* and wt mice.

each week (Fig 3C) identified two genes with significant differences at every week: *Chordc1* and *Psat1* (Fig 3D). Interestingly, the profiles of *Chordc1* showed almost complete separation between WT and *mdx* mice at every week.

Out of the 1532 genes differentially expressed in *mdx* mice, 181 were found to be also differentially expressed in *mdx++* and *mdx+−* mice (Dataset EV3, Tabs 2_WT_mdx++ and 3_WT_mdx+−). Eight of these genes (*Cd59a, Chordc1, Hpgd, Ldhb, Lincred1, Myo6, Slc39a11,* and *Tlr1*) displayed significant differences at each week both in the WT vs *mdx++* and in the WT vs *mdx+−* comparisons (Fig 3E). Comparison of the logFCs in the *mdx++* and *mdx+−* groups showed a strong correlation at every week (Fig 3F), indicating a high level of similarity between the two mouse groups.

Pathway analysis with IPA showed clear differences related to mice age. At 6 weeks, the elevation of the sirtuin signaling pathway was in line with the observation in muscle; synaptogenesis signaling, consistent with a phase of high muscle regeneration, was also identified. A clear inflammatory profile was visible at this young age, which normalized at week 12 and was depleted at week 18. Upstream regulator analysis in IPA showed that genes such as TGF-β and PGR (but also drugs such as ST1926 and simvastatin) could explain the observed signature (Fig 3H). There were no strong associations at the later time points. Examples of genes showing concordant expression in blood and muscle were *Fermt3, Hpse, Cd44,* and *F13a1*.

Not surprisingly, there was not a clear overlap in pathways affected in muscle and blood, as genes expressed in blood are largely different than the ones expressed in muscle tissue. To understand if and how gene expression in blood can be used to remotely monitor gene expression in muscle, we proceeded to integrate the muscle and blood expression data.

**Integration of the blood and muscle signatures**

Given the availability of paired muscle and blood RNA-seq data at the 30 week time point, we investigated how much of the muscle gene expression signature is traceable in blood. Integration of muscle (MoMus dataset) and blood (MoLong dataset) data was performed by comparing the correlation networks in muscle and blood inferred through WGCNA. WGCNA identified 15 modules of co-expressed genes in muscle (Dataset EV4, Tab 1_WGCNA_muscle_modules), and 19 modules in blood (Dataset EV4, Tab 2_WGCNA_blood_modules). The percentage of genes previously identified as differentially expressed between WT and *mdx* mice varied largely across muscle modules (Fig 4A), where it ranged from 1.6% (midnightblue module) to 70.4% (brown module) and showed also substantial variability across blood modules (Fig 4B), where it ranged from 3.6% (salmon module) to 34% (green module).

Assessment of the similarity of modules in the two networks was performed by looking both at the overlap of genes between blood and muscle modules (using the Jaccard index and overlap coefficient), and at the distribution of links between and within modules in the two networks (CSV indices). The Jaccard index between pairs of muscle and blood modules ranged between 0% and 12.7% (Appendix Fig S1A), indicating an overall low / moderate level of overlap. However, as highlighted by the overlap coefficient, a few module pairs were such that one module was largely contained in the other (Appendix Fig S1B): in particular, the pink muscle module shared 60.9% of its genes with the black blood module. Finally, the values of the CSV indices were found to be rather high (relative CSV index for the blood modules in the muscle network = 66%; relative CSV index for the muscle communities in the blood network: 78%), indicating that despite the only moderate overlap between muscle and blood modules, genes within each muscle module were connected rather tightly also in the blood network, and genes within each blood module were tightly connected also in the muscle network.

To identify cross-correlations between blood and muscle modules, a correlation analysis was performed on the extracted first PC of each module (the so-called "module eigengene") across the two tissues. A few blood modules were found to be highly correlated to two blocks of muscle modules (Fig 4C): The green and magenta blood modules displayed strong positive correlations with the yellow, pink, and black muscle modules, and strong negative correlations with the tan and magenta muscle modules. The cyan and greenyellow blood modules displayed negative correlations with the yellow, and positive correlation with the tan and magenta muscle modules.

The computation of correlations between blood module hubs and muscle eigengenes further showed that in several instances, the expression of a single gene in blood correlated with the expression of a module of genes in muscle as summarized by the module eigengene (see Dataset EV5, Tab 1_blood_green + magenta and Tab 2_blood_greenyellow + cyan for a few examples). These correlations indicate the possibility that a few genes in blood might be monitored to obtain information over the expression of groups of co-expressed genes in muscle. In particular, the expression of *Atp5a1* in blood significantly correlated with the expression of genes in the magenta muscle module, the expression of *Cnep1r1* correlated with genes in the yellow, tan, and magenta muscle modules, and the expression of *H2-Ob* correlated with the expression of genes belonging to the tan and magenta muscle modules.

To understand what muscle processes could be monitored by analyzing the expression of *Atp5a1, H2-Ob,* and *Cnep1r1* genes in blood, we looked into the pathways enriched in magenta, yellow, and tan modules. For example, the analysis of the muscle module yellow by Panther GO-Slim biological processes clarified that

    

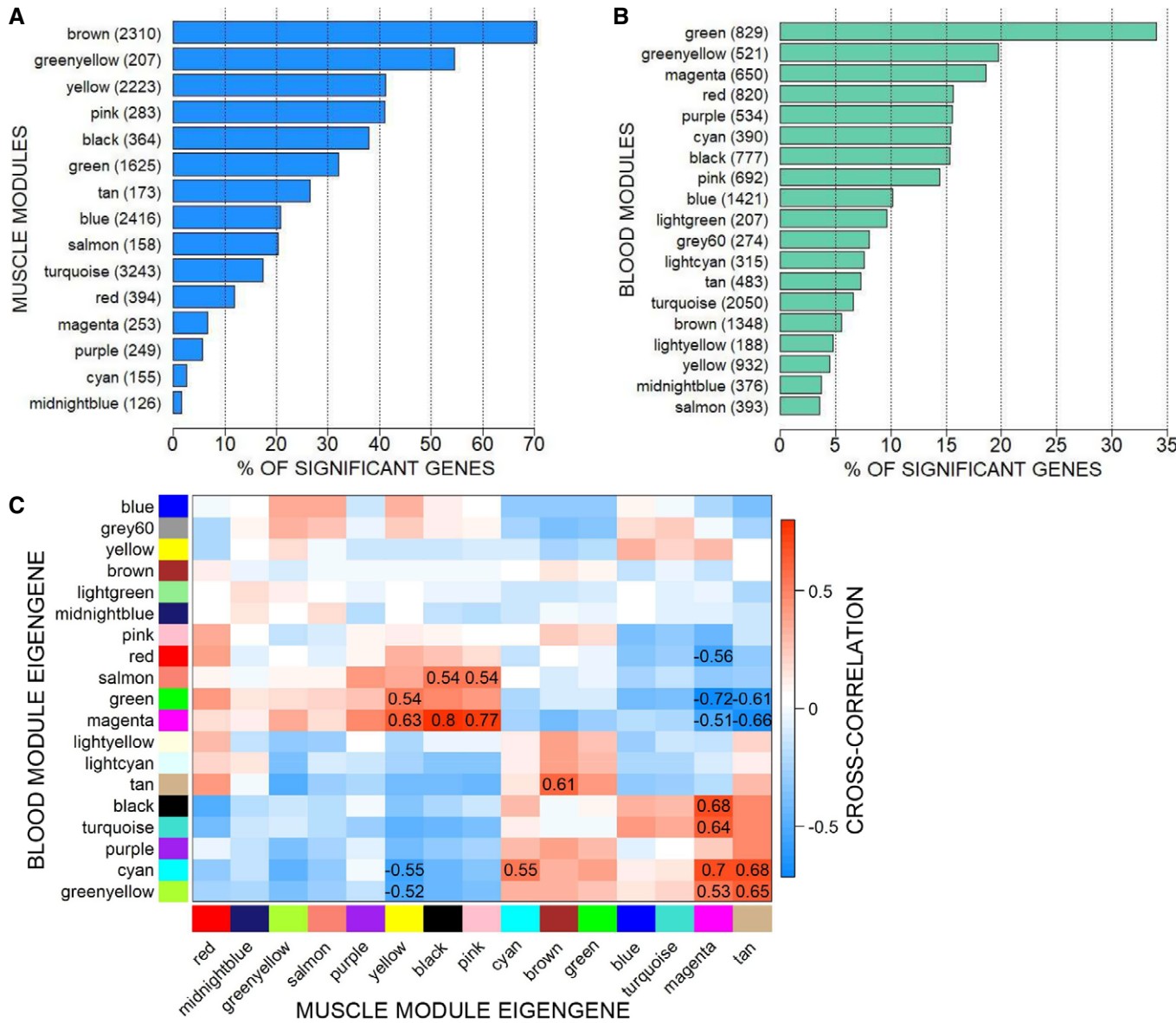

**Figure 4. Integration of the MoMus and MoLong datasets.**

A  Barplot showing the percentage of significant genes in the muscle modules. Next to each module name, we report the total number of genes that it comprises.
B  Barplot showing the percentage of significant genes in the blood modules. Next to each module name, we report the total number of genes that it comprises.
C  Matrix with the values of the cross-correlation between blood and muscle module eigengenes. To enhance visibility, only correlations with absolute value above 0.5 are shown.

genes in this module referred to the inflammatory and immune response component of the disease, consistent with T-cell activation and chemokine and cytokine signaling as evidenced by Panther over-representation pathway analysis. Interestingly, factors such as TGF-β1, CSF2, and IFN-γ are known upstream regulators of this signature. Therefore, expression levels of *Cnep1r1* in blood provide information on the inflammatory and immune component in muscle. Given that *Atp5a1* is primarily expressed in muscle, while *H2-Ob* and *Cnep1r1* are not, it is likely that changes in *Atp5a1* expression are the result of primary effects on muscle, while changes in *H2-Ob* and *Cnep1r1* are more likely

secondary changes less related to the muscle condition. Cross-correlation for hub genes across blood and muscle modules is reported in Dataset EV6.

**Treatment with PMO produced partial normalization of *mdx* mice toward WT, while PS49 further worsened *mdx* gene expression**

Given that DMD is caused by absence of dystrophin, multiple strategies aiming to restore dystrophin in muscle have been developed and tested both in preclinical experiments and in clinical trials, with

variable results (Verhaart & Aartsma-Rus, 2019). One of the challenges has been the inability to objectively evaluate the potentially beneficial effects of dystrophin restoration using non-invasive procedures. To address this, we used 2 antisense oligonucleotides that were previously reported to restore dystrophin in *mdx* mice. The selected oligonucleotides had chemistries used in clinical trials and were 2'-O-methyl phosphorothioate (PS49 oligo) and phosphorodiamidate morpholino oligomer (PMO). Dosing was evaluated to obtain dystrophin levels comparable with dystrophin recovery observed in recent clinical trials. Therapeutic exon skipping was observed for treated mice (Fig 5A), with average exon skipping of 5% for PS49 and 60% for PMO (Dataset EV7). Dystrophin

restoration quantified by Western blot averaged 0.23% for PS49, and 7.23% for PMO (Fig 5B).

Assessment of the effect of the PS49 and PMO treatments on gene expression levels in blood samples was performed considering 395 genes that were found to be differentially expressed at week 12 in *mdx* mice in the analysis of the MoLong dataset. For 127 genes, the effect of antisense treatment was concordant for both drugs and in line with the desired treatment effect, which is given by the opposite of the logFC observed in *mdx* (Appendix Fig S2).

Hypothesis testing led to the identification of 14 genes with significant differences (FDR < 0.05) between the 3 treatment groups (Dataset EV8, Tab 1_omnibus test). Differences between the PS49

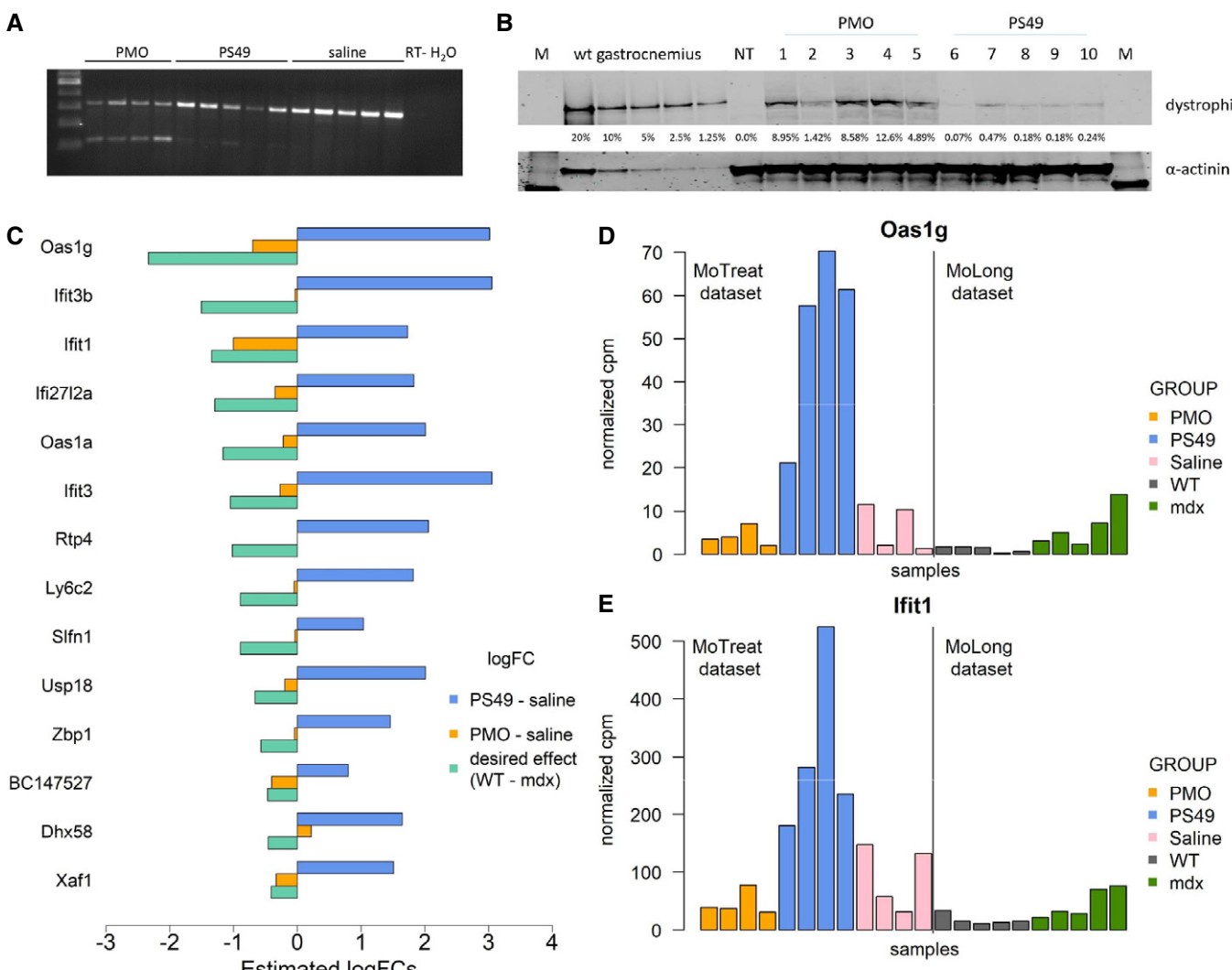

**Figure 5. Analysis of the effect of dystrophin restoration by antisense oligonucleotides on gene expression in blood (MoTreat dataset).**

A    RT–PCR (30 cycles) showing skipping of exon 23 for PMO and PS49 oligos. No skipping was visible in saline-treated mice.

B    Western blot showing the percentage of restoration of dystrophin protein in *mdx* mice following antisense oligonucleotides administration. M is molecular ladder. WT gastrocnemius represents the standard curve using protein extracts from gastrocnemius muscle. NT is saline-treated *mdx* mice.

C    Barplot comparing the estimated logFC for the PS49 and PMO treatments in the MoTreat dataset to the desired drug effect obtained from the MoLong dataset (logFC WT vs. mdx at week 12).

D, E    Histograms comparing the distribution of selected genes across different mice groups (PMO, PS49, and saline in the MoTreat dataset, WT and mdx at week 12 in the MoLong dataset).

and placebo groups were significant for all these 14 genes (Dataset EV8, Tab 12_PS49 vs placebo), while differences between PMO and saline groups were never significant (Dataset EV8, Tab 3_PMO vs placebo). More importantly, the logFCs associated with PS49 treatment showed exacerbation of the signature, indicating possible side effects related to the backbone chemistry (Fig 5C). Differentially expressed genes mapped especially to the interferon response. On the contrary, PMO showed a trend toward restoration of the signature, possibly in line with the larger dystrophin restoration by PMO. The detrimental effect of PS49 and the beneficial effect of PMO were particularly evident from the levels of *Oas1g* (Fig 5D) and *Ifit1* (Fig 5E), which could be considered candidate biomarkers for dystrophin-restoring therapies with antisense oligonucleotides.

## Assessment of differential expression and of the effect of steroid treatment in DMD patients

To assess whether the strong genetic signature identified in blood from dystrophic mice was also present in DMD patients, we studied a cohort of 39 DMD patients (of whom 32 treated with corticosteroids) and 18 age-matched healthy controls (Table 2). PCA showed separation between healthy individuals and DMD patients, and a certain overlap between patients treated with corticosteroids and untreated ones (Fig 6A).

A first hypothesis test for differences between the healthy, DMD-treated, and DMD-untreated groups led to the identification of 6657 genes significantly associated with the DMD group and steroid treatment (Dataset EV9, tab 1_global_test). In particular, 5589 genes were differentially expressed in DMD patients (Fig 6B and Dataset EV9, tab 2_test_group). *MYOM2* was the gene with the highest estimated fold change up to 9 years of age, but this logFC significantly decreased over time. Interestingly, DE genes were affected by age, while non-DE genes were more stable; typically, upregulated genes decreased with age, while downregulated ones tended to increase (Fig 6C). A total of 103 differentially expressed genes are coding for proteins previously reported in the analysis of serum of DMD patients and healthy controls (Spitali *et al*, 2018). Examples of the consistent directional change are shown in Appendix Fig S3.

Comparison of the lists of genes DE in *mdx* mice and in DMD patients led to the identification of 688 genes for which evidence of differential expression is present for both species (Dataset EV9, tab 3_overlap_with_mouse). Overall, we identified more genes as differentially expressed in DMD patients than in *mdx* mice, and 45%

of the 1532 genes identified as DE in *mdx* mice were also DE in DMD patients.

We further assessed the overlap between this list of differentially expressed genes and two previous microarray studies. The first study compared DMD patients and healthy controls aged 3 through 20 using an Affymetrix whole-genome human U133 Plus 2.0 GeneChips microarray (Wong *et al*, 2009), and the second compared DMD patients and healthy controls aged 3 through 10 using an Affymetrix GeneChip Human Gene 1.0 ST microarray (Liu *et al*, 2015). We found a moderate overlap between our study and that of Wong *et al* (2009), confirming 520 (50.5%) of the 1030 genes identified as differentially expressed in their study, and a high overlap between our study and that of Liu *et al* (2015), confirming 62 (73.8%) of their 84 significant genes (Fig 6D). Moreover, 11 genes were identified as differentially expressed in all 3 studies: *ATP5MPL*, *CD4*, *CYFIP1*, *DUSP6*, *EPB41L3*, *GAS5*, *GATA2*, *HRH4*, *NLRP3*, *PID1*, and *RPL13*.

Treatment with steroids affected the expression of 18 genes, among which 11 were upregulated and 7 downregulated (Fig 6E and Dataset EV9, tab 4_test_treatment). Ten of these 18 genes (*CEACAM6, CEACAM8, DEFA4, FCRL3, FCRL5, LCN2, MMP8, OLFM4, and OLR1*) had previously been linked to treatment with steroids by Liu *et al* (2015) (Fig 6F). Interestingly, the estimated treatment effect was largely opposite to the group effect, supporting a beneficial effect of the drug (Fig 6G). Steroid effect was particularly clear for *COL17A1* and *ADAMTS2* (Appendix Fig S4), the latter previously reported to show expression changes following exposure to these drugs (Hofer *et al*, 2008). Pathway analysis showed reduced oxidative metabolism as observed for the analysis of skeletal muscle in mice, providing further evidence that some muscle-related pathways can be monitored in blood. The overlap with pathways identified in mouse blood showed that the 18 week time point was the most similar to the observations in DMD patients (Fig 6H).

To assess whether gene expression in blood can be used to infer information on patients' clinical performance, we tested whether associations exist with the set of body measurements and performance tests listed in Dataset EV1. We found all body measurements to have strong positive correlations with each other (Appendix Fig S5A) and therefore proceeded to summarize them with their first principal component (PC), which explained 70% of their cumulative variance; this component can be interpreted as an overall body measurement index (Appendix Fig S5B). We tested the association of this PC with gene expression and identified 100 genes significantly associated with this body index, supporting the use of these

---

**Figure 6.  Results of the analysis of the DMDMex dataset.**

A   Principal component analysis of the DMDMex RNA-seq counts.

B   Volcano plot for the F test on differential expression between healthy individuals and DMD patients.

C   Scatter plot showing the group effect at baseline (x-axis) and the effect of the interaction between age and group (y-axis). Black dots represent non-differentially expressed genes, while blue dots represent differentially expressed genes.

D   Venn diagram showing the overlap between the genes identified as differentially expressed in DMD patients in Wong *et al* (2009), in Liu *et al* (2015) and in this study.

E   Volcano plot for the F test on differential expression between treated and untreated DMD patients.

F   Barplot comparing the logFC of the genes associated with steroid treatment estimated in this study to the estimates of Liu *et al* (2015).

G   Scatter plot showing genes for which a significant effect of treatment with steroids. The group effects at baseline are plotted on the x-axis, while the effects of steroids are plotted on the y-axis.

H   Heatmap obtained by Ingenuity Pathway Analysis comparing the pathways affected in DMD patients to the pathways affected at different time points in mouse.

I   Correlation circle representing the correlation between physical tests and their first two principal components.

J   Scatter plot showing the relationship between the first principal component of body measurements and the expression levels of LAPTM4B.

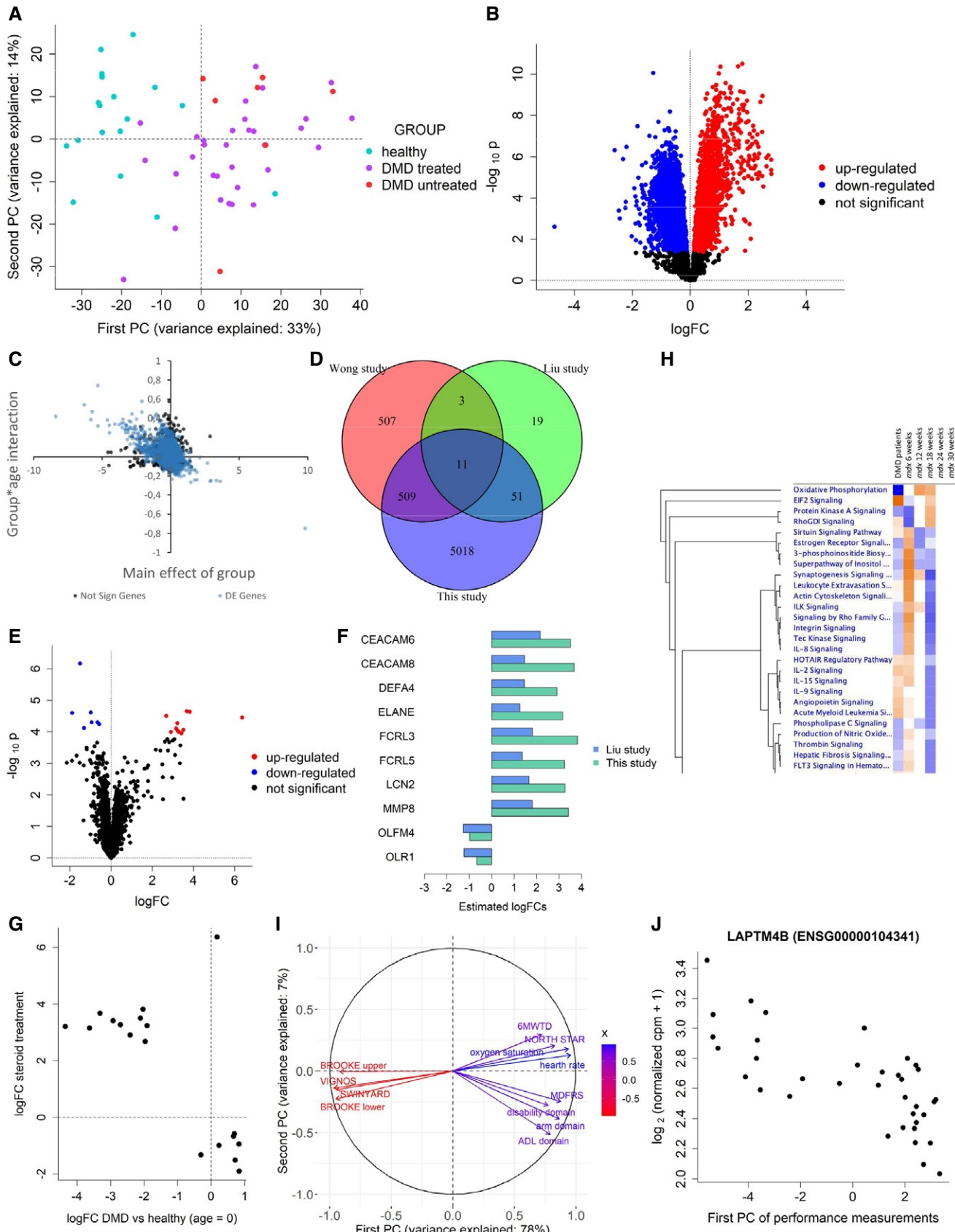

**Figure 6.**

markers as growth trackers in DMD patients (Appendix Fig S5C and Dataset EV10, tab 1_body_measurements).

A correlation analysis of the physical performance tests data led to the identification of two blocks of highly correlated tests (Appendix Fig S6A). The first block comprised the BROOKE (upper and lower), VIGNOS, and SWINYARD scales; these are indicators of disease severity, i.e., they assume higher values for patients with a more advanced stage of the disease. The second block comprised the six minute walk test distance, the NORTH STAR ambulatory assessment, the mobility domain (MDFRS scale), the activities of daily living domain (ADL) scale, disability domain, arm domain, heart rate, and oxygen saturation; all these variables are indicators of, or correlate with, the level of ambulation of a patient, and thus they tend to assume higher values for patients with a less advanced stage of the disease. These two blocks of variables furthermore displayed strong negative correlations with each other, indicating that to a good extent they provide complementary information. Given these strong correlations, we summarized the physical test measurements with their first PC, which explained 78% of their cumulative variance (Fig 6I) and it can be interpreted as an indicator of the physical condition of the patient (the higher it is, the better the patient's physical condition). A test on the association of this PC with gene expression led to the identification of one significant association (Appendix Fig S6B and Dataset EV10, tab 2_physical_tests) with *LAPTM4B*, which was elevated in patients at a more severe stage of the disease (ENSG00000104341, Fig 6J and Supplementary Fig 6C–F).

# Discussion

DMD is a rare genetic condition affecting males, for which no cure is available. Several therapeutic strategies failed during clinical development, and only a few have received conditional or accelerated approval by European, Japanese, and US regulatory agencies. One of the major obstacles in clinical development is the lack of objective readouts able to non-invasively describe the status of skeletal muscle during short-lived clinical trials. Typically, for dystrophin-restoring treatments investigators need to obtain baseline and follow-up muscle biopsies to assess drug effects. While such analyses are indeed useful to show target engagement, more in-depth analyses are required to prove consistent and long-lasting changes in muscle biology, which could anticipate improvement in patient performance. Therefore, the availability of peripheral biomarkers tracking altered pathways in the musculature would enable drug developers to improve their understanding of the effect of complex biologics by non-invasive blood sampling. A number of studies reported identification of different serum biomarkers (protein, miRNA, and metabolites) being elevated or reduced in patients compared to healthy controls. However, attempts to relate the observation in blood to the muscle biology have been limited and focused on single time points.

In this study, we used RNA-seq in muscle to have an in-depth description of the disease biology in 3 dystrophic mouse models; we further used the same technique to assess how to trace the muscle signature in blood over time. Analysis of muscle tissue showed how the lack of dystrophin severely affects the metabolic capacity of muscles with reduced oxidative phosphorylation, TCA cycle, and fatty acid β-oxidation; a concomitant increase in the inflammatory

profile was visible, as well as a potentially compensatory activation of the sirtuin signaling pathway. This signature was extremely conserved between the 3 animal models. No significant differences in gene expression were found between *mdx* utrophin+/+ and *mdx* utrophin+/− mice, where some mild phenotypic differences were previously reported (van Putten *et al*, 2012). Analysis of blood samples showed that a large number of genes were differentially expressed in dystrophic mice especially at 6 but also at 12 weeks of age, matching the disease phase characterized by intensive muscle regeneration. Two genes, *Chordc1* and *Psat1*, were differentially expressed at all time points, with *Chodc1* showing a complete separation between healthy and dystrophic mice, therefore showing promising attributes to monitor the disease over the whole period. The largest overlap in affected pathways between muscle and blood samples was visible at 6 weeks of age with inflammatory pathways (e.g., leukocyte extravasation signaling), but also the sirtuin pathway. An inversion of the directional changes was observed at week 18, marking a shift in how disease progresses from this time point onwards, and the end of the phase of intense muscle regeneration. As the disease further stabilized toward weeks 24 and 30, the number of differentially expressed genes dropped with consequent lack of significant pathway enrichment, making it challenging to compare the signature of paired muscle and blood samples at 30 weeks.

To enable a comparison between blood and muscle samples, we proceeded to build networks of co-expressed genes in both muscle and blood, and we used the first PC of each module to summarize the expression pattern of co-expressed genes. This enabled us to identify genes in blood that correlate with modules of co-expressed genes in muscle, suggesting their use as remote sensors for ongoing muscle processes. An example is represented by the expression levels of *Atp5a1* and *Cnepr1* in blood that correlated with the expression of the magenta and yellow muscle modules. Compared to *Cnep1r1*, *Atp5a1* is more likely to deliver muscle-specific information, since the primary source of expression of *Atp5a1* is muscle, while *Cnep1r1* is expressed in several tissues. The strength of this approach relies on the ability to summarize specific muscle pathways by monitoring the expression of a few genes in blood; however, in our case this was limited by the number of mice involved and the low signal at 30 weeks in blood.

Given the substantial signal at 12 weeks of age, we decided to assess whether dystrophin restoration by means of oligonucleotide-mediated exon skipping was able to normalize gene expression in blood, therefore providing a blood biomarker for dystrophin-restoring therapies. Two different antisense oligonucleotides were carefully dosed in order to obtain dystrophin recovery fitting the range of dystrophin restoration observed in recent clinical trials.

Treatment with PS49 oligo with the 2'-*O*-methyl nucleotides and phosphorothioate backbone showed < 1% dystrophin recovery and an exacerbation of the inflammatory profile, especially interferon response with genes such as *Ifit1*, *Ifit*3 *Ifit3b*, *Ifi27l2a*, *Oas1a*, and *Oas1g*, probably due to the phosphorothioate backbone dependent activation of the human toll-like receptor 9 (TLR9) response, despite the presence of the modification at the 2' position. Interestingly, the expression of 2'-5' oligoadenylate synthetases (such as *Oas1a* and *Oas1g* in our case) and interferon-induced protein with tetratricopeptide repeats (such as *Ifit1* and *Ifit3* in our case) were reported to be elevated after intracerebroventricular administration of oligonucleotides with the same

modifications, and after subcutaneous injection of 2'-methoxyethyl modification suggesting effects related to these chemistries (Toonen *et al,* 2018a; McCabe *et al,* 2020).

Treatment with PMO restored higher levels of dystrophin in muscle, consistent with the dystrophin recovery observed in recent trials with high dose of PMO (Clemens *et al,* 2020). No significant effects on interferon response were observed, and directional changes were in line with PS49 for the majority of the genes. Despite the lack of formal significance, treatment with oligonucleotides caused directional changes toward WT levels for 127 genes. Genes of the complement cascade showed reduced fold change for both drugs. A reduction in decorin and biglycan was also observed for both drugs (in line with collagen gene *Col3a1* and *Col1a1* reduction), previously reported to be elevated in patients biopsies and perhaps indicating a reduction in the fibrotic pathways following dystrophin restoration (Fadic *et al,* 2006). Interestingly, a reduction of *Igfbp6,* recently reported also in a phase 1b study with Rimeporide (Previtali *et al,* 2020), was observed; this reduction may hint at an increase of circulating free Igf1 levels, which are known triggers of muscle growth.

Analysis of gene expression in blood from DMD patients showed an even stronger signature than in mice and showed the potential to monitor pathways known to be affected in muscle such as, for example, reduced oxidative phosphorylation. The observed signature was more comparable to the 18 week time point in mice, probably indicating the limited regenerative capacity in patients, which is observed in the early time points in mice and reduced in 18 weeks old *mdx* mice. A clear steroid signature was observed with increase of known response markers such as *ADAMTS2* (Hofer *et al,* 2008). Interestingly, our results exhibited directional changes consistent with previous proteomics studies (Spitali *et al,* 2018), supporting the use of gene expression to monitor association previously discovered using other omic technologies. Lastly, *LAPTM4B* was significantly associated with patient performance, and patients with higher *LAPTM4B* expression showed a worse performance in physical tests. The *LAPTM4B* gene has been found to be associated with cell proliferation, invasion, and metastasis in cancer, by blocking lysosomal degradation and by protecting cells from autophagy (Tan *et al,* 2015). Moreover, it has been reported to promote activation of the mammalian target of rapamycin 1 (mTOR—inducing muscle growth) (Milkereit *et al,* 2015) and to reduce TGF-β production in human regulatory T cells (Huygens *et al,* 2015), therefore supporting the association identified in our analysis. The strength of the signature observed in blood from DMD patients is particularly remarkable if compared to similar studies that attempted to find a blood signature in milder dystrophies through RNA-seq (Signorelli *et al,* 2020b), but found very little evidence of differential expression.

In conclusion, this study provides evidence that analysis of gene expression in blood of preclinical and clinical samples of dystrophinopathies can deliver information of pathways affected in skeletal muscle, and it can be employed to show response to dystrophin-restoring therapy (even after sub-optimal dystrophin recovery) and to monitor patient performance. We expect that these findings will enrich the clinical trial toolkit for drug developers working on DMD.

# Materials and Methods

### Overview of the datasets described in the study

The data analyzed in this article were collected during two experiments performed at the Leiden University Medical Center (LUMC) in Leiden, NL, which involved healthy and dystrophic mice, and in an observational study carried out at the Sociedad Mexicana de la Distrofia Muscular AC (SMDM) in Mexico City, MX, which involved DMD patients and healthy subjects (Fig 1). Table 1 summarizes the features of the datasets described in the article. All blood samples of murine and human origin included in this study were whole blood samples including cellular RNA. This study did not focus on the evaluation of cell-free nucleic acids.

### Experiment 1 (MoMus and MoLong datasets)

The first experiment performed at the LUMC aimed at comparing gene expression profiles in healthy and dystrophic mice (Fig 1A). Four different mouse groups were included in the experiment: healthy wild-type (WT) mice, *mdx* mice, and *mdx* mice carrying 1 or 2 functional copies of the paralog gene utrophin (respectively abbreviated as *mdx +−* and *mdx ++* hereafter). WT and *mdx* mice shared the same genetic background (C57BL/10ScSn), while *mdx +−* and *mdx ++* had a mixed background. Both *mdx* and *mdx ++* mice had 2 functional utrophin copies, so the genetic background was the only difference between these 2 groups. Nine healthy and 31 dystrophic mice were housed in individually ventilated cages. Blood samples were obtained at 5 different time points. Before sampling, mice were fasted for 4–6 h, but free access to water was maintained. The samples at 6, 12, 18, and 24 weeks of age were obtained via an incision in the tail vein; the last blood sample was obtained via the eye at week 30. After sampling at 30 weeks, mice were sacrificed by cervical dislocation. Blood samples were obtained in RNeasy Protect Animal Blood Tubes (Qiagen, cat. N. 76544).

The experiment led to the collection of two datasets: a cross-sectional dataset, hereafter referred to as MoMus, that consisted of 40 tibialis anterior muscles, 31 collected from mice aged 30 weeks (7 WT, 8 *mdx,* 8 *mdx++,* 8 *mdx+−*), and 9 at earlier time points (for

**Table 1. Overview of the datasets described in this study.**

| Study and location | Dataset name | Species | Tissue | Dataset type | Number of subjects | Number of samples | Groups |
|---|---|---|---|---|---|---|---|
| Experiment 1, LUMC | MoMus | Mouse | Muscle | Cross-sectional | 40 | 40 | WT, *mdx, mdx++, mdx+−* |
| Experiment 1, LUMC | MoLong | Mouse | Blood | Longitudinal | 20 | 90 | WT, *mdx, mdx++, mdx+−* |
| Experiment 2, LUMC | MoTreat | Mouse | Blood | Cross-sectional | 14 | 14 | 3 *mdx* treatment groups: saline, PMO, PS49 |
| Observational study, SMDM | DMDMex | Human | Blood | Cross-sectional | 57 | 57 | DMD, healthy patients |

mice that died before week 30); and a longitudinal dataset, hereafter referred to as MoLong, that comprised 90 longitudinal blood samples collected at weeks 6, 12, 18, 24, and 30 from 20 of the 40 mice included in the experiment (5 WT, 5 *mdx*, 5 *mdx++*, and 5 *mdx+−* mice). The experiment was approved by the local animal welfare committee (DEC number 13154).

### Experiment 2 (MoTreat dataset)

The second experiment performed at the LUMC was designed to study the effect that dystrophin restoration produced by two anti-sense oligonucleotides has on blood gene expression (Fig 1B). Fourteen *mdx* mice with genetic background C57BL/10ScSn entered the experiment at 4 weeks of age. Mice were injected intravenously with either 200 mg/kg of a 2-O'-methyl antisense oligonucleotide with a phosphorothioate backbone (PS49) (WO2009054725A2—Means and methods for counteracting muscle disorders—Google Patents), or with 100 mg/kg of a phosphorodiamidate morpholino oligomer (PMO)(Heemskerk *et al*, 2009), or with saline solution. Mice were injected once per week for a period of 8 weeks; the total volume injected was 100 μl. Blood samples were obtained via the tail vein one week after the last injection, when mice were 12 weeks of age. Mice were sacrificed after blood collection. The results of the experiment were collected in a dataset, hereafter referred to as MoTreat, with serum RNA-seq samples from 14 *mdx* mice belonging to three treatment groups: saline, PMO, and PS49. Exon skipping was assessed in quadriceps from treated mice by RT–PCR. RNA was retrotranscribed with Bioscript reverse transcriptase (cat. n. BIO-27036, Bioline, London, UK) using 1 μg of total RNA and random heaxamers. Amplification was performed for 30 cycles with m22f (ATCCAGCAGTCAGAAAGCAAA) and m24r (CAGCCATC CATTTCTGTAAGG) primers. Quantification of the skipped product was performed using the Agilent DNA 1000 kit (cat n. 5067-1504) on the Agilent Bioanalyzer (Agilent, Santa Clara, US). Dystrophin quantification was performed by Western blot using recombinant anti-dystrophin antibody (ab154168, Abcam, diluted 1:2000) and IRDye 680TL goat-anti-rabbit as previously described (Jirka *et al*, 2018). Alpha-actinin (ab72592, Abcam, diluted 1:1000) was used as loading control. The experiment was approved by the local animal welfare body, and it was registered as AVD1160020171407.

### Observational study (DMDMex dataset)

The observational study performed at the SMDM involved patients with a confirmed DMD diagnosis based on clinical and genetic testing. Patients came from different states of Mexico; at the day of the evaluation, they were asked to fast before blood sampling. Blood was obtained in PAXgene® Blood RNA tubes. RNA purification was performed using the PAXgene Blood RNA Kit (Qiagen, Cat. N. 762174). Patients were allowed to eat breakfast and rest before performing physical tests. Trained medical doctors performed the tests with the help of physiotherapists as part of the routine evaluation for monitoring steroid treatment. The study was approved by the Centro Médico Nacional 20 de Noviembre in Mexico City (approval number 397.2014). The investigation was conducted according to the declaration of Helsinki and the principles set out in the Department of Health and Human Services Belmont Report.

This observational study led to the collection of serum samples from 39 DMD patients and from 18 healthy controls (Fig 1C). Hereafter, we refer to this dataset as DMDMex. Age and RNA-seq expression levels in blood samples were available for all subjects; furthermore, for DMD patients, further information on treatment with steroids, body measurements, and performance tests was collected (Table 2). The list of the available body measurements and physical tests can be found in Dataset EV1.

### RNA-sequencing

#### Murine samples

RNA samples were purified using the RNeasy Protect Animal Blood Kit (Qiagen, Cat. N. 73224). Globin mRNA content was reduced using the GLOBINclea Kit for mouse/rat (Thermo Fisher, Cat. N. AM1981) following the manufacturer's instructions. RNA quality was confirmed using the LabChip GX 96-well RNA kit (Perkin Elmer). Successful globin depletion was assessed by qPCR for both Hba and Hbb mRNAs. Sample preparation and RNA-seq were performed at deCODE genetics (Reykjavik, Iceland). Sample preparation was performed using the TruSeq Poly-A v2 Kit (Illumina, San Diego). Quality of the sequencing libraries was assessed by sequencing indexed pooled samples on Miseq (Illumina, San Diego), while sequencing was performed on HiSeq 2500 (2 × 125 cycles) as previously described (Toonen *et al*, 2018b).

#### Human samples

Depletion of globin mRNA content was performed using the human GLOBINclear Kit (Thermo Fisher, Cat. N. AM1980); RT–qPCR on both HBA and HBB transcripts was performed to confirm reduction of globin expression following depletion. Sample preparation and sequencing were performed at GenomeScan (Leiden, NL). Sample concentration was determined using the Fragment Analyzer. The Illumina Truseq RNA sample prep kit v2 (RS122-2001) was used to process the samples. Briefly, mRNA was selected using poly-A beads. Adapter ligation followed by amplification by PCR was performed. The samples were analyzed on the Fragment Analyzer. Clustering and DNA sequencing using the Illumina NextSeq 500 was performed according to manufacturer's protocols.

### RNA-seq alignment, filtering, and normalization

Sequencing data were analyzed using the BIOPET Gentrap in-house pipeline (https://biopet-docs.readthedocs.io/en/stable/pipelines/gentrap/). Quality control was performed using FastQC and MultiQC. Murine data were aligned to mouse reference genome GRCm38 using STAR aligner version 2.3.0e with an average of 81.6% alignment ratio. The gene read quantification was performed using HTSeq-count (v0.6.1) and the UCSC mm10 gene annotation, where an average of 59% of reads can be assigned uniquely to known genes. Human data were aligned to human reference genome GRCh38 using GSNAP version 2017-09-11 with an average of 93.5% alignment ratio. The gene read quantification was performed using HTSeq-count (v0.6.1) and the Ensembl gene annotation version 87, where an average of 76% of reads can be assigned uniquely to known genes.

Pre-processing of RNA-seq counts was performed separately for each of the datasets described in Table 1. We removed lowly expressed genes from each dataset by considering only genes with at least 5 counts per million (cpm) in at least 10% of the samples. RNA-seq expression profiles were normalized separately in each

cohort using the Trimmed Mean of M values (TMM) normalization method (Robinson & Oshlack, 2010).

## Statistical analysis of the MoMus dataset

Statistical modeling of the cross-sectional MoMus dataset was performed using the limma-voom pipeline (Law *et al*, 2014) on RNA-seq counts from the 31 biopsies obtained at week 30. Mouse group was included as covariate in the limma-voom model, and the F test was employed to test differential expression between WT and *mdx* mice, as well as between WT and *mdx*++, WT and *mdx*+−, and *mdx*++ and *mdx*+− mice. Results were corrected for multiple testing using the Benjamini–Hochberg method (Benjamini & Hochberg, 1995). Pathway analysis for this dataset, as well as for the MoLong and DMDMex datasets, was performed using Ingenuity Pathway Analysis (IPA).

## Statistical analysis of the MoLong dataset

Statistical modeling of the longitudinal MoLong dataset was performed using linear mixed models (McCulloch *et al*, 2008) with precision weights estimated from voom (Law *et al*, 2014). We first identified genes differentially expressed between WT and *mdx* mice considering a linear mixed model (LMM) that included a mouse-specific random intercept, and time (categorical) and group (WT, *mdx*) as fixed-effect covariates. A likelihood ratio test (LRT) was used to identify differentially expressed genes between WT and *mdx* mice. Results were corrected for multiple testing using the Benjamini–Hochberg method (Benjamini & Hochberg, 1995). For genes with significant differences, we further tested differences at each specific time point using the Wald test. Moreover, for the top genes of the WT versus *mdx* analysis we considered a LMM in combination with the voom precision weights where time (categorical) and group (WT, *mdx*++, *mdx*+−) were included as fixed effects. A LRT was used to identify differentially expressed genes between WT and *mdx*++, WT and *mdx*+−, and *mdx*++ and *mdx*+− mice.

## Statistical integration of the MoMus and MoLong datasets

Integration of blood and muscle expression profiles was carried out using the Weighted Gene Co-expression Network Analysis (WGCNA) method (Langfelder & Horvath, 2008). For this analysis, we applied a more stringent filtering threshold, retaining genes with at least 10 cpm in at least 40% of the samples. Muscle and blood

samples obtained at week 30 were normalized with the TMM method and used to estimate two tissue-specific co-expression networks. Node clustering was applied to each network, deriving modules of co-expressed genes for which module eigengenes were computed. The overlap between blood and muscle modules was assessed considering the Jaccard index and the overlap coefficient. Assessment of the similarity of the structure of the blood and muscle networks in terms of distribution of links between and within modules was carried out using the Community Structure Validation (CSV) index (Cutillo & Signorelli, 2018). Cross-correlations between blood and muscle module eigengenes were computed using Pearson's correlation coefficient, using matched samples across the two tissues. Panther GO-Slim and Panther over-representation pathway analysis (Mi *et al*, 2012) were used to characterize genes belonging to the WGCNA modules.

## Statistical analysis of the MoTreat dataset

Comparison of gene expression profiles across treatment groups in the MoTreat dataset focused on a set of 416 genes identified in the MoLong dataset as differentially expressed between WT and *mdx* mice at week 12. Twenty-one of those genes were excluded from the analysis because they did not pass the filtering threshold. Two mice were excluded from the analysis because the quality of the RNA-sequencing was found to be sub-optimal, with a very low percentage (31% and 28%) of correctly aligned reads. Statistical analysis of the data was carried out using the edgeR pipeline (Robinson *et al*, 2010). We considered negative binomial (NB) generalized linear models (GLMs) where the expression levels of each gene depend on treatment group (placebo, PMO, PS49). Differences between groups were tested using the quasi-likelihood F (QLF) test. Results were corrected for multiple testing using the Benjamini–Hochberg method.

## Statistical analysis of the DMDMex dataset

Statistical modeling of the DMDMex dataset was performed using edgeR (Robinson *et al*, 2010). Hypothesis testing was based on the QLF test. Results were corrected for multiple testing using the Benjamini–Hochberg method.

With the aim of identifying genes differentially expressed between healthy individuals and DMD patients, we fitted a NB GLM with age, group, their interaction, and treatment as covariates. We first computed an omnibus test that assessed whether a gene was either different between healthy and DMD subjects, or between treated and untreated patients. For the genes identified as significant (FDR < 0.05) by this test, we proceeded to identify genes differentially expressed between healthy and DMD subjects. Moreover, we identified genes differentially expressed between treated and untreated patients.

For the DMD patients, we further studied association between gene expression levels and the body measurements and physical tests listed in Dataset EV1. Given the large number of available covariates, each block of variables was summarized through principal component (PC) analysis. The first PC of the body measurements explained 70% of the total variability of the 24 measurements, whereas the first PC of the physical tests explained 78% of the variability of the 12 tests. Identification of the genes associated with the body measurements was carried out using a NB

**Table 2. Overview of the variables available in the DMDMex dataset.**

| Group of individuals | Number of individuals | Average age (min; max) | Available variables |
|---|---|---|---|
| DMD patients | 39 | 9.1 (4; 31) | RNA-seq expression in blood; age, treatment with steroids (binary), body measurements, and performance tests |
| Healthy controls | 18 | 11.2 (5; 13) | RNA-seq expression in blood; age |

**The paper explained**

**Problem**
Duchenne muscular dystrophy (DMD) is a rare neuromuscular disorder characterized by progressive muscle degeneration, loss of motor skills, and premature death. The development of therapeutic approaches has been delayed by difficulties to monitor safety and efficacy in clinical trials in a non-invasive manner. To facilitate the drug development process, more knowledge connecting the biology of the disease with clinical progression and the ability to objectively monitor clinically relevant changes in a non-invasive manner are needed.

**Results**
In this study, we characterize the evolution over time of gene expression in blood of dystrophic mice and show how dystrophic changes in muscle are reflected in blood by analyzing paired muscle and blood samples. We identify genes whose expression in blood is associated with the safety and efficacy of treatment with two different antisense drugs that yield different levels of dystrophin restoration. Furthermore, we compare blood gene expression in DMD patients and in healthy individuals, showing overlap and differences between mouse and man. We also show how gene expression responds to treatment with corticosteroids.

**Impact**
The results presented in this study provide evidence that blood RNA-sequencing can serve as a tool to evaluate disease progression in dystrophic mice and patients, as well as to monitor response to (dystrophin-restoring) therapies in preclinical drug development and in clinical trials.

GLM where we included the first PC of the body measurements as covariate; similarly, identification of genes associated with the physical tests was performed with a NB GLM where we included the first PC of the physical tests as covariate.

## Data availability

The datasets produced in this study are available in the following databases:

1    Mouse RNA-seq data: Gene Expression Omnibus repository, accession id GSE132741 (https://www.ncbi.nlm.nih.gov/geo/query/acc.cgi?acc=GSE132741)
2    Human RNA-seq data: European Genome-phenome Archive, accession id EGAS00001004907 (https://www.ebi.ac.uk/ega/studies/EGAS00001004907).

**Expanded View** for this article is available online.

## Acknowledgements

We kindly acknowledge funding from the *Duchenne Parent Project NL* foundation, the *Association Française contre les myopathies* (grant number 19118), the *Spieren voor Spieren* foundation (grant number Svs15), and the *European Commission* through the project *Neuromics* (grant number 305121) and the *RD-Connect* platform (grant number 305444). We thank Kees Fluiters for running the pathway analyses with Ingenuity.

## Author contributions

Plan and implementation of statistical analyses; interpretation of results; manuscript writing; and preparation of figures, tables, and supplementary materials: MS. Plan and implementation of statistical analyses; and manuscript revision: ME and OV. Plan statistical analyses; and manuscript revision: KH. RNA-sequencing of human data; and manuscript editing: NV. Execution of mice experiments; and manuscript editing: RG and CLT. Collection of clinical data; and manuscript editing: LBLH, REC, and BGD. RNA-sequencing of mice data; and manuscript revision: OTM. RNA-sequencing of mice data; and manuscript editing: HM. Plan and implementation of statistical analyses; interpretation of results; and manuscript editing: RT. Interpretation of results; and manuscript editing: AA. Execution of mice experiments; planning of statistical analyses; interpretation of results; manuscript writing; and preparation of figures, tables, and supplementary materials: PS.

## Conflict of interest

The authors declare that they have no conflict of interest.

## For more information

Research group websites:
- Website of the DMD Genetic Therapy Group of LUMC http://www.exonskipping.nl/
- Duchenne Centrum Nederland: https://duchenneexpertisecentrum.nl/research/
- Integrated European Project on Omics Research of Rare Neuromuscular and Neurodegenerative Diseases https://rd-neuromics.eu/
- RD-Connect project https://rd-connect.eu/

Patient associations:
- AFM Telethon: http://www.afm-telethon.com/
- Duchenne Parent Project (NL) https://duchenne.nl/
- Duchenne Parent Project: https://www.parentprojectmd.org/
- Duchenne Parent Project (IT) http://parentproject.it/
- Muscular Dystrophy Association https://www.mda.org/
- Cure Duchenne (US) https://www.cureduchenne.org/
- Fight Duchenne Foundation (AU) http://www.fightduchenne.org.au/
- Fight DMD (US) http://www.fightdmd.com/

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
