## [Review Process File · EMBO Molecular Medicine]

Peripheral blood transcriptome profiling enables monitoring disease progression in dystrophic mice and patients

Mirko Signorelli, Mitra Ebrahimipoor, Olga Veth, Kristina Hettne, Nisha Verwey, Raquel García-Rodríguez, Christa Tanganyika-deWinter, Luz Berenice Lopez Hernandez, Rosa Escobar Cedillo, Benjamín Gómez Díaz, Olafur Magnusson, Hailiang Mei, Roula Tsonaka, Annemieke Aartsma-Rus, and Pietro Spitali

DOI: [10.15252/emmm.202013328](https://doi.org/10.15252/emmm.202013328)

Corresponding authors: *Mirko Signorelli (m.signorelli@lumc.nl)* , *Pietro Spitali (P.Spitali@lumc.nl)*

Review Timeline:

Submission Date:	24th Aug 20
Editorial Decision:	30th Sep 20
Revision Received:	8th Dec 20
Editorial Decision:	15th Jan 21
Revision Received:	9th Feb 21
Accepted:	10th Feb 21

Editor: Zeljko Durdevic

Transaction Report:

30th Sep 2020

Dear Dr. Signorelli,

Thank you for the submission of your manuscript to EMBO Molecular Medicine. We have now received feedback from the three reviewers who agreed to evaluate your manuscript. As you will see from the reports below, the referees acknowledge the interest and novelty of the study but also raise some concerns that should be addressed in a major revision. Particular attention should be given to the independent validation of the candidate biomarkers as well as to more detailed analysis of the datasets produced in this study and comparison to already existing datasets.

Addressing the reviewers' concerns in full will be necessary for further considering the manuscript in our journal, and acceptance of the manuscript will entail a second round of review. EMBO Molecular Medicine encourages a single round of revision only and therefore, acceptance or rejection of the manuscript will depend on the completeness of your responses included in the next, final version of the manuscript. For this reason, and to save you from any frustrations in the end, I would strongly advise against returning an incomplete revision.

We realize that the current situation is exceptional on the account of the COVID-19/SARS-CoV-2 pandemic. Therefore, please let us know if you need more than six months to revise the manuscript.

I look forward to receiving your revised manuscript.

Yours sincerely,

Zeljko Durdevic

***** Reviewer's comments *****

Referee #1 (Comments on Novelty/Model System for Author):

The mouse model is known to be milder than the human counterpart. The authors are aware of this

Referee #1 (Remarks for Author):

The study by Signorelli compares gene expression signatures between muscle and blood taken from the mdx mouse model of Duchenne muscular dystrophy. The goal of the study is to establish better biomarkers for DMD, including those that reflect treatment status. A number of studies have

evaluated protein and metabolic markers using materials from both animal models and human samples. This study collects blood and muscle from the same animals over time and compares these to animals treated with antisense oligonucleotides, and then additionally compares the results to blood samples from humans with DMD. The concept is a good one since reliable, treatment responsive biomarkers are needed. However, there are elements of the approach and analysis that trigger a number of questions.

1. It would be helpful to understand which genes share expression patterns between blood cells and muscle. For example, *Atp5a1* is expressed highly in skeletal muscle and is also expressed highly in lymphocytes. Does the shift in this gene occur as a primary or secondary deficit from loss of dystrophin. In contrast *Cnep1r1* has comparatively low expression in muscle and thus the change in expression might more accurately reflect primary changes in blood. The results and discussion around specific genes and pathways should more carefully inform the readers whether the authors consider these primary or secondary deficits. This may be especially important since the human samples are derived from many DMD patients who are on steroids, where this does not seem to be the case for the mice. The use of steroids could more directly be impacting the expression.

2. The change in metabolism gene expression is also quite interesting. However it is unclear the degree to which these changes derive from the reduction in activity seen in both dystrophic animals and humans. The authors should consider comparing to other data sets to get a sense whether the gene expression changes are primary related to the pathology in DMD or whether they reflect reduced activity (and perhaps loss of muscle mass). Can the data be normalized to mass?

3. A more careful comparison to previously published proteomic or metabolomic studies should also be added. While gene expression in the peripheral blood cells is potentially powerful, the relatively ease of directly measuring protein or metabolites seems more likely to translate to a useful biomarker. Thus, a more integrated evaluation as to how these data compare with other proteomic/metabolomic studies should be included.

Minor

1. The processing of blood samples would appear to be evaluating RNA from the blood as opposed to an evaluation of cell free nucleic acid. This should be explicitly stated if this is correct. Also in the human samples, are these results altered by the percent neutrophils vs lymphocytes?

Referee #2 (Remarks for Author):

The manuscript by Signorelli et al provides interesting data on transcriptomic changes in muscle and blood of 3 different dystrophic mouse models over a period of seven months. They then extended their analysis to evaluate the effect of two AONs and identify genes associated with different outcomes. They then conclude their studying blood gene expression in a large cohort of DMD patients.

The authors address an important point, as there is a critical need to identify novel reliable biomarkers to monitor disease progression and efficacy of new interventions in DMD (creatin kinase is useful, but has limitations).

The study is essentially descriptive but very well designed, with solid data in multiple models,

timepoints and species. Statistical analysis also appears appropriate, although the appropriateness of some methods is difficult for me to assess as it goes beyond my expertise (I'm not a bioinformatician).

The main issue I have with this study is the difficulty in filtering take home messages and critical genes which the authors would like to suggest the field to use as biomarkers, as I have a feeling that this essential information at times gets diluted in the very many technical details of the bioinformatic analyses. Also, it might be unfeasible to perform RNAseq analyses routinely in daily clinical neuromuscular practice, hence it might be useful to see evidence that some of those genes/markers could be easily monitored with cheaper and simpler assays.

Referee #3 (Remarks for Author):

In this manuscript, the authors set to uncover blood biomarkers that can monitor the pathophysiological changes occurring in skeletal muscle from dystrophic versus healthy subjects (both mice and humans). To this end, the authors employed RNA-seq analysis in mice and paired blood and muscle samples in order to characterize the dystrophic signature over five different time points. Furthermore, the authors evaluated the effect of two different antisense oligo drugs on dystrophin restoration and compared the molecular signature to that seen in WT versus mdx mice. Lastly, a human study was performed in parallel with the mouse studies, in which blood from DMD patients (either treated with corticosteroids or untreated) and healthy controls were subjected to comparative RNA-seq analysis. While the main concept of the study is compelling, and several new candidate biomarkers were described, there are several major points of criticism for the manuscript as well as multiple minor points:

Major:

1. In general, the authors failed to independently validate expression of candidate biomarkers in either the blood or the muscle in mdx versus WT mice. Examples of candidates that were mentioned based on the RNA-seq dataset are *Prune2*, *Chordc1*, *Psat1*, *Oas1g*, and *Ifit1*. While the identification of these genes being differentially expressed is exciting, validation by qPCR is needed to further describe their involvement in pathophysiology.
2. There was no comparison of the differential genes found in the mouse study to the genes found in the human study. One of the weaknesses in the field of biomarker identification in muscular dystrophy lies in the inconsistencies between mouse and human studies. In the current study, you have taken on both murine and human datasets, so the opportunity to compare the two are present, yet there is no attempt at this. If there are same genes that are significantly changed between dystrophic and healthy subjects in both your mouse and human studies, it would greatly validate the use of that gene as a robust biomarker.
3. It is unclear why the differences in several signaling pathways in mdx compared to WT are lost as the age of the mice used increases (Figure 3G and H). The authors describe the lack of differences in the older age groups as a "stabilization of the disease". If that is the case, the differences seen at 18 weeks of age should be unchanged (and therefore still different between WT and mdx).
4. One parameter in the study between WT and mdx mice that is perhaps more important in terms of pathophysiology than age is exercise. This study would be remarkably strengthened by RNA-seq of blood isolated from WT and mdx at 18 weeks of age that undergo an exercise regime versus remaining sedentary. This dataset could then be compared to the list of differentially expressed genes found in patients based on several body measurements.

Minor:

1. There is confusion in the definition of the "3 different dystrophic mouse models". The authors

assigned "mdx ++" to mdx mice that have both functional alleles of utrophin, and "mdx+-" for mdx mice with only one functional allele of utrophin, but I am wondering if there is a difference between "mdx" and "mdx++"? The standard mdx mouse strain indeed has both functional alleles of utrophin, unless these "mdx" mice are actually "mdx/utr-/-" DKO mice? Or is the difference solely a mixed versus congenic background? This information needs to be explained in the Methods section and nomenclature within the paper should be adjusted.

2. The authors mentioned that there have been phenotypic differences found in mdx++ versus mdx+- mice, but yet in Figure 1G they do not detect any differentially expressed genes between these two groups of mice. However, the volcano plot indicates that a few genes seem to trend as significant. What parameters made these genes come up as "non-significant"?

3. Figure 1K is missing a key for the color coding of expression.

4. In Figure 4, it would be nice to have specific genes within the modules discussed in the Results section in the main figure.

5. Figure 5C shows the 14 genes that were identified via hypothesis testing as downregulated in mdx muscle versus WT, then compared to expression after treatment with both PS49 and PMO antisense drugs. Were no upregulated genes identified in this testing?

6. The authors compared the significant changes in gene expression found in their human study with two other previous studies as a Venn diagram in Figure 6D. In this panel, 11 genes were found to be consistent among the three different studies. It would be very important to list these genes and to further validate them via qPCR independently. Also, the authors claim that their study and the Liu study have "very high overlap" even though only 62 of the 5,589 genes found in this study were also found in the Liu study.

REPLY TO REVIEWER # 1

Overview: The study by Signorelli compares gene expression signatures between muscle and blood taken from the mdx mouse model of Duchenne muscular dystrophy. The goal of the study is to establish better biomarkers for DMD, including those that reflect treatment status. A number of studies have evaluated protein and metabolic markers using materials from both animal models and human samples. This study collects blood and muscle from the same animals over time and compares these to animals treated with antisense oligonucleotides, and then additionally compares the results to blood samples from humans with DMD. The concept is a good one since reliable, treatment responsive biomarkers are needed. However, there are elements of the approach and analysis that trigger a number of questions.

Reply: We thank the reviewer for the positive feedback and the constructive comments, which we answer hereafter.

Major comment #1: It would be helpful to understand which genes share expression patterns between blood cells and muscle. For example, *Atp5a1* is expressed highly in skeletal muscle and is also expressed highly in lymphocytes. Does the shift in this gene occur as a primary or secondary deficit from loss of dystrophin? In contrast *Cnep1r1* has comparatively low expression in muscle and thus the change in expression might more accurately reflect primary changes in blood. The results and discussion around specific genes and pathways should more carefully inform the readers whether the authors consider these primary or secondary deficits. This may be especially important since the human samples are derived from many DMD patients who are on steroids, where this does not seem to be the case for the mice. The use of steroids could more directly be impacting the expression.

Reply: This question refers to the cross-correlation analysis of WGCNA module eigengenes presented in Figure 4, where we connect the paired muscle-blood RNA-seq data obtained at the 30 weeks-time point. In the manuscript we show how some module pairs are particularly correlated, and we mention *Atp5a1*, *Cnep1r1* and *H2-Ob* as interesting genes in blood. The interesting feature of these blood hub-genes is that they correlate with the module eigengene of certain muscle modules, meaning that the expression of these genes in blood at week 30 relates to the expression of a number of interconnected genes in muscle belonging to the magenta, tan and yellow modules.

Given that the blood hub genes are expressed at different levels in muscle and blood, the reviewer asks whether a possible shift in expression in these 3 hub-genes in blood is more likely caused by changes in gene expression in blood or muscle. To answer this question we looked at the expression patterns for these genes in the EMBL expression atlas, Illumina body map, Human protein atlas and Gtex databases (see image below for an overview).

As mentioned by the reviewer, *Atp5a1* expression is indeed higher in muscle compared to blood, while the expression of *H2-Ob* and *Cnep1r1* is more evenly distributed across tissues, or more prevalent in blood (see figure below for both mouse and human). Therefore, it

appears likely that changes in *Atp5a1* in blood are more strongly connected to changes in gene expression in muscle as the reviewer suggests. We have added a mention of this fact both in the Results and in the Discussion sections.

Mouse

Atp5a1

H2-Ob

Cnep1r1

Human

ATP5F1A

HLA-DOB

CNEP1R1

Major comment #2: The change in metabolism gene expression is also quite interesting. However it is unclear the degree to which these changes derive from the reduction in activity seen in both dystrophic animals and humans. The authors should consider comparing to other data sets to get a sense whether the gene expression changes are primary related to the pathology in DMD or whether they reflect reduced activity (and perhaps loss of muscle mass). Can the data be normalized to mass?

Reply: The reviewer here asks a great question, which we can partially answer with the data available. This question is about the data presented in Figure 2, where we show the results of the analysis of gene expression in muscle.

As a first point, please note that in our experiments we did not measure mouse activity, since the mice were not in metabolic cages and they did not receive training or performance assessment by rotarod, treadmill or hanging tests. It is therefore difficult to directly relate the gene expression changes in metabolic pathways found in muscle RNA-seq to activity.

There have been, however, studies relating genes to activity. To compare our results to such studies, we have gathered the list of differentially expressed genes mapping to the metabolic pathways heavily altered in dystrophic mice, namely: Mitochondrial Dysfunction, Sirtuin Signaling Pathway, Oxidative Phosphorylation and TCA Cycle II (Eukaryotic). This produced a list of 115 differentially expressed genes. We checked whether these genes were previously reported in publications associated with physical activity via the EURETOS knowledge platform using the human gene identifiers, as this is a richer information subset in terms of associations compared to the mouse gene IDs. We also included other concepts such as reduced muscle, mitochondria, metabolic aspects and sirtuins as control associations that we expected to find back.

	physical activity	duchenne	"sirtuins"	"metabolic aspects"	"mitochondria"	"muscle"	"muscular dystrophy"
PPARA (homo sapiens)	0	0	170	47	5111	4881	40
PPARG (homo sapiens)	0	0	112	25	3207	2810	21
CS (homo sapiens)	0	0	8	4	2081	2124	12
SOD2 (homo sapiens)	0	0	80	8	3176	641	6
CASP3 (homo sapiens)	0	0	11	8	3230	461	4
JUN (homo sapiens)	0	0	19	9	1893	1598	5
PARP1 (homo sapiens)	0	0	113	9	3048	312	3
UCP2 (homo sapiens)	0	0	13	3	1909	502	0
AIFM1 (homo sapiens)	0	0	9	1	2019	99	0
SUCLA2 (homo sapiens)	0	0	4	11	428	1621	13
BCL2 (homo sapiens)	0	0	8	2	1531	269	6
DLD (homo sapiens)	0	0	1	1	733	920	7
VDAC1 (homo sapiens)	0	0	0	4	1252	116	2
BECN1 (homo sapiens)	0	0	15	2	1015	281	1
PPIF (homo sapiens)	0	0	7	0	736	50	3

As expected, a large number of publications exist where the reported genes are associated with terms such as mitochondria, muscle as well as more specific terms such as sirtuins and muscular dystrophy. On the contrary, no strong associations were found between the genes and physical activity. While this does not constitute direct proof that no connection exists between the activity of the mice and gene expression, it is the most direct association that we can establish based on the data obtained in this study.

As concerns the idea of normalizing the gene expression data to mass: while the relation between mass and metabolism is interesting, including mass in a model designed to test

differential expression in mdx mice with respect to controls would be problematic. This is due to the fact that dystrophic mice are consistently heavier compared to WT mice: this induces a strong association between the WT / mdx status and mass, which would produce a multicollinearity problem in the estimation of a regression model that would include both mass and group as covariates.

Major comment #3: A more careful comparison to previously published proteomic or metabolomic studies should also be added. While gene expression in the peripheral blood cells is potentially powerful, the relatively ease of directly measuring protein or metabolites seems more likely to translate to a useful biomarker. Thus, a more integrated evaluation as to how these data compare with other proteomic/metabolomic studies should be included.

Reply: we agree with the reviewer that integrating the peripheral blood signature with metabolomic and proteomic data is important.

Comparison between RNA-seq and proteomic data is more straightforward compared to integration with metabolic (and lipidomic) data, as gene and protein ID are directly comparable.

We therefore compared the 1532 differentially expressed genes found in blood with a list of proteins recently identified by a wide screening in 14 weeks old *mdx* mice (Coenen-Stass et al., 2015). This study found 96 proteins identified by aptamer-based serum proteomics. A total of 11 genes/proteins were identified in both studies. These are: C1QBP, CYCS, DNAJC19, FYN, LDHB, PCNA, PGD, PRKACA, PTPN11, TYMS and UFM1.

Comparison of the fold changes (FCs) showed that the direction of the change was generally concordant between the RNA-seq data (at week 12, and somewhat more at week 18) and the proteomics data (mice were 14 weeks old in the paper by Coenen-Strass et al). Concordant changes were observed for C1QBP, CYCS, DNAJC19, LDHB, PCNA, PGD and UFM1. The discordant directional changes for PRKACA and PTPN11 were perhaps caused by earlier changes in gene expression, as the directional change is concordant with the directional changes observed at week 6 (see figure below).

We then compared the magnitude of the changes. LogFCs were higher for these proteins in the proteomic dataset. While the extremes in terms of logFC are somewhat higher in the proteomics screening, in general the logFCs in the RNAseq dataset are in the same order of magnitude (see figure below).

Integration of the RNA-seq data presented in this manuscript with metabolomic data is a more challenging question, but an interesting one, which we are currently researching. Our goal is that of integrating the peripheral blood data from the MoLong dataset (this article) with metabolomic (Tsonaka et al., 2020) and lipidomic data (article under review) that were collected alongside with the blood transcriptomics.

This is ongoing work, which we plan to include in a future manuscript that will focus on the integration of the different omic sources. Although this integration falls beyond the scope of the present manuscript, hereafter we would like to share some preliminary results that we have obtained applying Multiomic Factor Analysis (MOFA) to blood RNA-seq, metabolomic and lipidomic data.

MOFA (Argelaguet et al., 2018) is a method for the integration of multiomic data that leads to the identification of latent factors that can be shared across different omic sources, but also specific of a single source. Estimation of MOFA led us to identify 9 latent factors (LFs) that together explain 77.8% of the total variance of blood RNA-seq counts, 30.7% of the variance

of metabolites and 14.9% of the variance of lipids. The table below provides a more detailed breakdown, with the percentage of variance explained by each LF in each omic view:

Factor	Blood RNAseq	lipids	metabolites
LF1	30.0%	2.2%	1.1%
LF2	2.9%	13.5%	5.6%
LF3	11.7%	5.3%	2.7%
LF4	14.0%	0.3%	0.1%
LF5	6.1%	4.8%	2.3%
LF6	2.4%	3.4%	0.1%
LF7	5.7%	0.0%	0.0%
LF8	2.6%	0.4%	2.7%
LF9	3.2%	0.0%	0.0%

Overall, we can observe that while some factors are almost unique to blood RNAseq (LF1, LF4, LF7, LF9), some other factors indicate the presence of a common signature between blood RNAseq, lipids and metabolites (LF2, LF3, LF5), or just between blood RNAseq and lipids (LF6), or between blood RNAseq and metabolites (LF8).

The results of the multiomic integration with MOFA will enable us to understand which genes, metabolites and lipids deliver similar or additional information on the status of the animals at the different time points. We will then be able to provide a more solid answer to the question raised by the reviewer.

Lastly, we have considered whether to add to the manuscript (part of) the additional results presented in this answer, especially considering that this rebuttal will be published alongside with the article. Our assessment is that such an addition would increase the length of the manuscript even further, and it would not help the reader in the interpretation. Therefore, we propose to leave this comparison out of the manuscript.

References:

1. Coenen-Stass, A. M., McClorey, G., Manzano, R., Betts, C. A., Blain, A., Saleh, A. F., ... & Roberts, T. C. (2015). Identification of novel, therapy-responsive protein biomarkers in a mouse model of Duchenne muscular dystrophy by aptamer-based serum proteomics. *Scientific reports*, 5, 17014.
2. Tsonaka, R., Signorelli, M., Sabir, E., Seyer, A., Hettne, K., Aartsma-Rus, A., & Spitali, P. (2020). Longitudinal metabolomic analysis of plasma enables modeling disease progression in Duchenne muscular dystrophy mouse models. *Human Molecular Genetics*, 29(5), 745-755.
3. Argelaguet, R., Velten, B., Arnol, D., Dietrich, S., Zenz, T., Marioni, J. C., ... & Stegle, O. (2018). Multi-Omics Factor Analysis—a framework for unsupervised integration of multi-omics data sets. *Molecular Systems Biology*, 14(6), e8124.

Minor comment #1: The processing of blood samples would appear to be evaluating RNA from the blood as opposed to an evaluation of cell free nucleic acid. This should be explicitly stated if this is correct. Also in the human samples, are these results altered by the percent neutrophils vs lymphocytes?

Reply: the reviewer is correct: we report analysis of whole blood, and there was no attempt to focus on cell free RNA. This has been clarified at the beginning of the Methods section with the sentence: "All blood samples of murine and human origin included in this study were whole blood samples including cellular RNA. This study did not focus on the evaluation of cell free nucleic acid."

Unfortunately, a direct answer to the question on the percentage of neutrophils vs lymphocytes is not possible, because a quantification of the number of neutrophils and lymphocytes present in blood was not performed. Nevertheless, we checked whether the proportion of lymphocytes vs neutrophils could be a confounder in the comparison of DMD patients and healthy controls by estimating the percentage of several cell types in each sample using the wbccPredictor (<https://github.com/mvaniterson/wbccPredictor>). In the Figure below we show that the distribution of the estimated percentage of lymphocytes (left panel) and of neutrophils (right panel) is very similar between healthy controls and DMD patients:

Thus, intuitively the estimated proportion of lymphocytes and neutrophils does not appear to be substantially different in the two groups. To verify this intuition, we computed the lymphocytes / neutrophils ratio, and tested the null hypothesis of no difference between the two groups. We obtained $p = 0.28$, a result that indicates that there is very little evidence that the proportion of lymphocytes and neutrophils differs between the two groups.

REPLY TO REVIEWER # 2

Overview: The manuscript by Signorelli et al provides interesting data on transcriptomic changes in muscle and blood of 3 different dystrophic mouse models over a period of seven months. They then extended their analysis to evaluate the effect of two AONs and identify genes associated with different outcomes. They then conclude studying blood gene expression in a large cohort of DMD patients.

The authors address an important point, as there is a critical need to identify novel reliable biomarkers to monitor disease progression and efficacy of new interventions in DMD (creatine kinase is useful, but has limitations).

The study is essentially descriptive but very well designed, with solid data in multiple models, timepoints and species. Statistical analysis also appears appropriate, although the appropriateness of some methods is difficult for me to assess as it goes beyond my expertise (I'm not a bioinformatician).

The main issue I have with this study is the difficulty in filtering take home messages and critical genes which the authors would like to suggest the field to use as biomarkers, as I have a feeling that this essential information at times gets diluted in the very many technical details of the bioinformatic analyses. Also, it might be unfeasible to perform RNAseq analyses routinely in daily clinical neuromuscular practice, hence it might be useful to see evidence that some of those genes/markers could be easily monitored with cheaper and simpler assays.

Reply: We thank the reviewer for the comment. It is indeed challenging to deliver a broad view of the data while focusing on a few key messages. We strived to provide examples of relevant genes identified across the different analyses, while providing access to the full results as supplementary materials. Moreover, in the new version of the manuscript we have tried to highlight the most important messages by putting more weight on the findings that can be more useful for other scientists and drug developers, such as (for example) the genes that are differentially expressed at all time points in blood, and the genes in blood that are more likely to monitor muscle specific changes (as also suggested by reviewer 1). We also highlighted the genes that could monitor the effects of therapy such as antisense in mouse and steroids in patients. We think that among the many genes identified as significant in our study, the genes that we have selected for presentation in the text and in the main figures are the ones that could be taken forward by other groups to monitor progression and response to therapy. We hope this new version is now clearer.

REPLY TO REVIEWER # 3

Overview: In this manuscript, the authors set to uncover blood biomarkers that can monitor the pathophysiological changes occurring in skeletal muscle from dystrophic versus healthy subjects (both mice and humans). To this end, the authors employed RNA-seq analysis in mice and paired blood and muscle samples in order to characterize the dystrophic signature over five different time points. Furthermore, the authors evaluated the effect of two different antisense oligo drugs on dystrophin restoration and compared the molecular signature to that seen in WT versus mdx mice. Lastly, a human study was performed in parallel with the mouse studies, in which blood from DMD patients (either treated with corticosteroids or untreated) and healthy controls were subjected to comparative RNA-seq analysis. While the main concept of the study is compelling, and several new candidate biomarkers were described, there are several major points of criticism for the manuscript as well as multiple minor points.

Reply: We thank the reviewer for the time spent to assess our manuscript, and for the valuable comments. Below we provide our answers to each of them.

Major comment #1: In general, the authors failed to independently validate expression of candidate biomarkers in either the blood or the muscle in mdx versus WT mice. Examples of candidates that were mentioned based of the RNA-seq dataset are *Prune2*, *Chordc1*, *Psat1*, *Oas1g*, and *Ifit1*. While the identification of these genes being differentially expressed is exciting, validation by qPCR is needed to further describe their involvement in pathophysiology.

Reply: The reviewer is indeed correct, we did not show validation experiments for the genes found to be differentially expressed. We acknowledge that qPCR is used as a method to validate sequencing associations; however, we did not proceed in this direction because the large number of differentially expressed genes identified in this study would have made it technically and financially not possible to approach the validation with qPCR.

In cases - such as this one - with a strong signature, instead of proceeding with a technical validation with a second technology, a biological validation is often proposed by analyzing other samples with the same technique. In this way, artifacts of a single association are often filtered out. In our case, associations found in muscle are often verifiable in datasets deposited in data repositories. In the example mentioned by the reviewer, *Prune2* was previously described here (<https://www.ncbi.nlm.nih.gov/geoprofiles/95738437>). Therefore, validation of the results obtained in muscle is often verifiable using deposited data where other techniques have been used to quantify gene expression.

Consulting repositories is less informative for blood gene expression, because no expression data for blood have been previously reported in mdx mice. The availability of repeated measurements (together with rigorous multiple testing correction) is however a good way to reduce the chance of false positives. In this case, *Chordc1* and *Psat1* were found to be

differentially expressed at 5 different time points, suggesting a clear association with the dystrophic phenotype and a negligible chance that these associations are due to technical issues.

The chance of false positives is however higher for genes differentially expressed at single time points. To address this point, we performed qPCR validation experiments for a selection of genes. We selected 3 genes (*H2-Eb2*, *Fbxo9*, *Mtss1*) that displayed global differences (adjusted p global < 0.05), and were furthermore differentially expressed at only 1 time point. This selection was made because genes that are DE at only 1 time point are more likely to be false positives.

In the table below we compare the results of the RNA-seq analysis to those obtained by qPCR at week 12:

Gene	RNA-seq data			qPCR validation	
	LogFC	Adj P global	P value 12 weeks	Mean Diff	P value 12 weeks
Mtss1	-0.569	0.027	1.053E-05	-0.101	0.056
H2-Eb2	-0.505	0.010	0.015	-0.053	0.038
Fbxo9	0.631	0.010	0.056	0.237	0.612

The analysis of the RNA-seq data led to the conclusion that the expression *H2-Eb2* and *Mtss1* at week 12 was lower in *mdx* mice compared to WT, while *Fbxo9* expression was elevated in *mdx* mice blood. qPCR analysis yielded a confirmation of these findings, identifying *Mtss1* and *H2-Eb2* as reduced in *mdx* mice, and *Fbxo9* as increased (compare logFC and Mean Diff columns).

Comparing the significance of the findings by RNA-seq data and qPCR is trickier, since the two techniques yield a different type of outcome (overdispersed counts vs continuous measurements) that are analysed with substantially different statistical models. Nevertheless, even when looking at significance we do see a good level of agreement between the two techniques: for *H2-Eb2*, the difference at week 12 was found to be significant both with RNA-seq and with qPCR. For *Mtss1* the difference was significant with RNA-seq, and trending to significant with qPCR. Lastly, the difference at week 12 for *Fbxo9* was not significant both with RNAseq and with qPCR.

The data presented above are meant to exemplify how the associations identified in our study could be reproduced with an independent technique. It is our belief that a more comprehensive validation by qPCR of all DE genes identified in our study is beyond the scope of this discovery work, and lies more in the analytical development of methods, which could be the topic of a follow-up study.

Major comment #2: There was no comparison of the differential genes found in the mouse study to the genes found in the human study. One of the weaknesses in the field of biomarker identification in muscular dystrophy lies in the inconsistencies between mouse and human studies. In the current study, you have taken on both murine and human datasets, so the opportunity to compare the two are present, yet there is no attempt at this. If there are same genes that are significantly changed between dystrophic and healthy subjects in both your mouse and human studies, it would greatly validate the use of that gene as a robust biomarker.

Reply: we thank the reviewer for this valuable comment. Indeed, in the previous version of the manuscript we checked the overlap between mouse and human only at the pathway level, and we did not include an assessment of the overlap at the gene level. However, the reviewer is absolutely right in pointing out that our data offer the opportunity to assess the consistency of findings across species. Therefore, we have now included a more thorough assessment of the overlap between genes found as DE in *mdx* mice and in DMD patients.

As mentioned in the Results section of the revised manuscript, comparison of the lists of DE genes in *mdx* mice and in DMD patients led to the identification of 688 genes for which evidence of differential expression is present for both species. Overall, we identified more genes as differentially expressed in DMD patients than in *mdx* mice, and 45% of the 1532 genes identified as DE in *mdx* mice were also DE in DMD patients. This is an important finding, that shows that despite the inconsistencies often found between mouse and human studies, it is possible to pinpoint to a good number of biomarkers that appear to be robust across species.

Major comment #3: It is unclear why the differences in several signaling pathways in mdx compared to WT are lost as the age of the mice used increases (Figure 3G and H). The authors describe the lack of differences in the older age groups as a "stabilization of the disease". If that is the case, the differences seen at 18 weeks of age should be unchanged (and therefore still different between WT and mdx).

Reply: this comment made us realize that our phrasing was ambiguous, since the reviewer interpreted what we previously called stabilization at weeks 24 and 30 as absence of changes compared to the previous time point. To remove this ambiguity and clarify what we meant, we proceeded to change the sentence in the discussion. The paragraph now reads:

“An inversion of the directional changes was observed at week 18, marking a shift in how disease progresses from this time point onwards, and the end of the phase of intense muscle regeneration”

Major comment #4: One parameter in the study between WT and mdx mice that is perhaps more important in terms of pathophysiology than age is exercise. This study would be remarkably strengthened by RNA-seq of blood isolated from WT and mdx at 18 weeks of age that undergo an exercise regime versus remaining sedentary. This dataset could then be compared to the list of differentially expressed genes found in patients based on several body measurements.

Reply: We appreciate the comment, and understand the interest of the reviewer for exercise in these mice. However, we do not fully understand what the reviewer has in mind when asking this comparison. First, patients can do only limited exercise. Exercise such as downhill treadmill can aggravate the phenotype of the *mdx* model, and perhaps the reviewer is interested in knowing what genes in blood are associated with a more severe phenotype. We tried to answer this question by including mice with different functional utrophin allele counts that were previously been described as differently affected. We also included multiple time points covering the more intensive phase (6 to 12 weeks) and the later time points.

Including a new arm of the study should be justified by a strong research question. It would be difficult to convince the welfare body to include a further arm in the study to compare exercised to unexercised mice without a clear a priori hypothesis. A comparison of exercised mice with the data already collected would furthermore suffer from batch effects, since the new batch of mice and sequencing data would belong to a different year and different litter (all reported data were obtained from mice that were included in the experiment at the same time point; mice from litters were randomized across groups to avoid litter effects). A further complication is that we have recently re-derived mice, therefore the results will suffer from extra complication due to re-derivation.

To conclude, we believe that to be able to assess the study of the effect of training on mdx mice we would need a completely new study setup, and we therefore kindly decline this request.

Minor comment #1: There is confusion in the definition of the "3 different dystrophic mouse models". The authors assigned "mdx ++" to mdx mice that have both functional alleles of utrophin, and "mdx+-" for mdx mice with only one functional allele of utrophin, but I am wondering if there is a difference between "mdx" and "mdx++"? The standard mdx mouse strain indeed has both functional alleles of utrophin, unless these "mdx" mice are actually "mdx/utr-/-" DKO mice? Or is the difference solely a mixed versus congenic background? This information needs to be explained in the Methods section and nomenclature within the paper should be adjusted.

Reply: We thank the reviewer for marking this point. We have edited the Methods and clarified that *mdx* and *mdx++* mice have no differences in terms of number of functional utrophin copies, and that the genetic background is the only difference between these 2 groups. When we designed the experiment, we reasoned whether to include only one of these 2 groups, as that would have saved power in the statistics and budget for the experiment. However, excluding the *mdx++* mice would have complicated the interpretation of potential differences between *mdx* and *mdx+-* mice as these have different genetic background. As for the nomenclature, we specify in the Methods what is intended for *mdx*, *mdx++* and *mdx+-*, and we refer to these abbreviations throughout the rest of the manuscript.

Minor comment #2: The authors mentioned that there have been phenotypic differences found in *mdx++* versus *mdx+-* mice, but yet in Figure 1G they do not detect any differentially expressed genes between these two groups of mice. However, the volcano plot indicates that a few genes seem to trend as significant. What parameters made these genes come up as "non-significant"?

Reply: throughout the article, we combined hypothesis testing with the Benjamini-Hochberg multiple testing correction, with the goal of minimizing the risk of generating false positives. Moreover, we followed the customary choice of setting $\alpha = 0.05$ as cutoff to classify a gene as DE (differentially expressed) or not DE. We have strived to be consistent with this approach throughout the whole article, since we believe that this is the most rigorous way to present our results, and that changing α across the different analyses based on the observed adjusted p-values would essentially correspond to p-hacking, and to cherry-picking when no significant genes are found.

Of course, choosing a threshold for differential expression is always somewhat arbitrary, and using a larger α would result in some genes being declared as differentially expressed between *mdx++* and *mdx+-*. However, since α corresponds to the probability of wrongly rejecting the null hypothesis when that hypothesis is true, choosing a larger α would increase the risk of producing false positive results.

Minor comment #3: Figure 1K is missing a key for the color coding of expression.

Reply: we are a bit unsure about this comment. The reasons of our confusion are that there is no Figure 1K in the manuscript, so we thought that maybe the reviewer was referring to Figure 2K. However, Figure 2K comprises a legend showing the color scale used in the image. It is possible that we misunderstood this comment; if this is the case, please let us know and we'd be happy to fix the issue.

Minor comment #4: In Figure 4, it would be nice to have specific genes within the modules discussed in the Results section in the main figure.

Reply: we have carefully pondered whether to take this suggestion into account, and modify the way in which we present the outputs of the WGCNA analysis. However, after making a few trials we felt that the resulting figure would be complicated and difficult to read, so we decided to keep Figure 4 as it was. Please note that this is the standard way in which WGCNA results are usually presented in publications.

Minor comment #5: Figure 5C shows the 14 genes that were identified via hypothesis testing as downregulated in *mdx* muscle versus WT, then compared to expression after treatment with both PS49 and PMO antisense drugs. Were no upregulated genes identified in this testing?

Reply: we feel that this comment might be due to a possible misunderstanding of the results presented in Figure 5C, so we would like to take the opportunity to clarify how Figure 5C was obtained.

For the analysis of the effect on gene expression of PS49 and PMO, we focused on the 395 genes that were differentially expressed in *mdx* mice blood at week 12 compared to WT mice. These genes were selected based on differences found in the blood comparison (not on muscle) as we wanted to investigate whether the drug effect can be monitored in blood. Among these 395 genes there are both upregulated and downregulated ones.

As we could not perform qPCR for 395 genes, we decided to sequence the RNA of antisense and saline treated mice, and then focus the analysis only on the 395 genes that we identified as differentially expressed in the comparison of WT and *mdx* mice (results presented in Figure 3). We therefore proceeded to test whether PMO and PS49 had an effect on the expression of the 395 genes in blood. Of the genes tested for differential expression across PMO and PS49 treatment groups, 14 were found to be significant. Coincidentally, these 14 genes are all upregulated in *mdx* blood at week 12.

Minor comment #6: The authors compared the significant changes in gene expression found in their human study with two other previous studies as a Venn diagram in Figure 6D. In this panel, 11 genes were found to be consistent among the three different studies. It would be very important to list these genes and to further validate them via qPCR independently. Also, the authors claim that their study and the Liu study have "very high overlap" even though only 62 of the 5,589 genes found in this study were also found in the Liu study.

Reply: as concerns the first point raised by the referee in this comment: the 11 genes found to be consistent across all three studies are ATP5MPL, CD4, CYFIP1, DUSP6, EPB41L3, GAS5, GATA2, HRH4, NLRP3, PID1 and RPL13. We have followed the suggestion and added their names to the manuscript.

With respect to the second point: the purpose of validating results by qPCR is to show that the results obtained are not an artifact due to the technique used. Technical validation is then followed normally by a biological/clinical validation where the scope is to assess whether the association found in one cohort is cohort-dependent (geographical effects - different standards of care – other). In this case, the 11 genes are found to be significant using different techniques already (microarray and RNAseq), therefore passing the first layer of technical validation required for a discovery paper such as this one. The validity of the association across different cohorts further supports the findings, as a second layer of biological validation (across cohort) is also satisfied. Assessing these genes by qRT-PCR would therefore not add to the findings, given that the association survive both technical and biological validation across different techniques and cohorts.

Lastly, we would like to clarify in what sense we see the overlap between the study from Liu and ours as "high". The study of Liu identified 84 genes as differentially expressed between DMD patients and healthy controls. Therefore, the maximum number of shared genes that can exist between our study and Liu's is 84. In practice, our study confirms 62 of the 84 genes found by Liu as differentially expressed in DMD, yielding an overlap coefficient equal to 73.8%. We consider an overlap coefficient of 73.8% to be high, especially considering that in this case gene expression was measured using different techniques (microarrays in Liu et al., RNA-seq in our manuscript).

15th Jan 2021

Dear Dr. Signorelli,

Thank you for the submission of your revised manuscript to EMBO Molecular Medicine. I am pleased to inform you that we will be able to accept your manuscript pending the following final amendments:

1) With the beginning of the new year, we encountered high number of submissions, so that our data editors were not able to process all received manuscripts. Therefore, we will send you the document with data editor's suggestions as soon as our data editors process your manuscript. Please do not submit your revised manuscript before we send you the file with data editor's suggestions. Thank you for your understanding.

2) Figures: Please upload individual, high-resolution figure files. For more information on figure presentation please check "Author Guidelines".

<https://www.embopress.org/page/journal/17574684/authorguide#datapresentationformat>

3) In the main manuscript file, please do the following:

- Correct/answer the track changes suggested by our data editors by working from the attached/uploaded document.
- Remove all figures from the text file and leave figure legends at the end of the text file.
- Remove text colour.
- Make sure that all special characters display well.
- Move M&M section after Discussion.
- In M&M, provide the antibody dilutions that were used for each antibody.
- In M&M, include a statement that informed consent was obtained from all human subjects and that, in addition to the WMA Declaration of Helsinki, the experiments conformed to the principles set out in the Department of Health and Human Services Belmont Report.
- Move Table 1 and 2 to the end of the manuscript.
- In "Author contributions" please use authors' initials instead of full names.
- In addition to the accession number please provide URL for the RNA sequencing data. Please be aware that all datasets should be made freely available upon acceptance, without restriction. Use the following format to report the accession number of your data:

[data type]: [full name of the resource] [accession number/identifier] ([doi or URL or identifiers.org/DATABASE:ACCESSION])

Please check "Author Guidelines" for more information.

<https://www.embopress.org/page/journal/17574684/authorguide#availabilityofpublishedmaterial>

- Correct the reference citation in the text and reference list. In the text of the manuscript, a reference should be cited by author and year of publication. Include a space between a word and the opening parenthesis of the reference that follows. In the reference list, citations should be listed in alphabetical order. Where there are more than 10 authors on a paper, 10 will be listed, followed by "et al.". Please check "Author Guidelines" for more information.

<https://www.embopress.org/page/journal/17574684/authorguide#referencesformat>

4) Appendix: The file with 6 suppl. figures should be renamed to Appendix, with table of content and figure legends. Correct nomenclature is "Appendix Figure S1" etc. Please check "Author Guidelines" for more information.

<https://www.embopress.org/page/journal/17574684/authorguide#expandedview>

5) Datasets: 10 suppl. files should be renamed to "Dataset EV1" etc. Each file should have a title and a short description within the file in a separate tab. Please also add callouts for datasets to the manuscript.

6) The Paper Explained: Please provide "The Paper Explained" and add it to the main manuscript text. Please check "Author Guidelines" for more information.

<https://www.embopress.org/page/journal/17574684/authorguide#researcharticleguide>

7) Synopsis: Every published paper now includes a 'Synopsis' to further enhance discoverability. Synopses are displayed on the journal webpage and are freely accessible to all readers. They include separate synopsis image and synopsis text.

- Synopsis image: Please provide a striking image or visual abstract as a high-resolution jpeg file 550 px-wide x (250-400)-px high to illustrate your article.

- Synopsis text: Please provide a short stand first (maximum of 300 characters, including space) as well as 2-5 one sentence bullet points that summarise the paper as a .doc file. Please write the bullet points to summarise the key NEW findings. They should be designed to be complementary to the abstract - i.e. not repeat the same text. We encourage inclusion of key acronyms and quantitative information (maximum of 30 words / bullet point). Please use the passive voice.

8) For more information: There is space at the end of each article to list relevant web links for further consultation by our readers. Could you identify some relevant ones and provide such information as well? Some examples are patient associations, relevant databases, OMIM/proteins/genes links, author's websites, etc...

9) As part of the EMBO Publications transparent editorial process initiative (see our Editorial at <http://embomolmed.embopress.org/content/2/9/329>), EMBO Molecular Medicine will publish online a Review Process File (RPF) to accompany accepted manuscripts. This file will be published in conjunction with your paper and will include the anonymous referee reports, your point-by-point response and all pertinent correspondence relating to the manuscript. Let us know whether you agree with the publication of the RPF and as here, if you want to remove or not any figures from it prior to publication. Please note that the Authors checklist will be published at the end of the RPF.

10) Please provide a point-by-point letter INCLUDING my comments as well as the reviewer's reports and your detailed responses (as Word file).

I look forward to reading a new revised version of your manuscript as soon as possible.

Yours sincerely,

Zeljko Durdevic

***** Reviewer's comments *****

Referee #1 (Comments on Novelty/Model System for Author):

Use of mouse models and comparison to human DMD patient.

Referee #1 (Remarks for Author):

The authors responded very nicely to the prior comments. The additional text and analyses improve the quality and interpretation of the findings.

Referee #3 (Remarks for Author):

In the resubmission of this manuscript, the authors have addressed every major and minor concern, which has led to clarity of the issues at hand and an overall strengthened manuscript. Changes have been made within the manuscript which include restating claims in a clearer manner as well as inclusion of the gene names of potential biomarkers found in cross-referenced datasets. The only minor concern that remains is the refusal to perform qPCR experiments to validate even just a few of the genes mentioned to be both differentially expressed and biologically interesting; however, the authors made a strong argument that many of the gene expression profiles of these genes can be found in online deposited datasets, and also that these genes have been cross-referenced to other existing datasets using another technique (i.e. RNA-seq here versus microarray data obtained elsewhere) which strengthens their findings. Therefore, the independent validation experiments is likely beyond the scope of this discovery-based paper.

The authors performed the requested editorial changes.

We are pleased to inform you that your manuscript is accepted for publication and is now being sent to our publisher to be included in the next available issue of EMBO Molecular Medicine.

Corresponding Author Name: Mirko Signorelli

Manuscript Number: EMM-2020-13328